# Box Thirding: Anytime Best Arm Identification under Insufficient Sampling

**Seohwa Hwang** [1]   **Junyong Park** [1]

## Abstract

We introduce Box Thirding (B3), a flexible and efficient algorithm for Best Arm Identification (BAI) under fixed budget constraints. It is designed for both anytime BAI and scenarios with large $N$, where the number of arms is too large for exhaustive evaluation within a limited budget $T$. The algorithm employs an iterative ternary comparison: in each iteration, three arms are compared—the best-performing arm is explored further, the median is deferred for future comparisons, and the weakest is discarded. Even without prior knowledge of $T$, B3 achieves an $\epsilon$-best arm misidentification probability comparable to SH, which requires T as a predefined parameter, applied to a randomly selected subset of $c_0$ arms that fit within the budget. Empirical results show that B3 outperforms existing methods for the limited budget constraint in terms of simple regret, as demonstrated on the New Yorker Cartoon Caption Contest dataset.

## 1. Introduction

In the standard multi-armed bandit (MAB) setting, a learner sequentially selects from a finite set of $N$ arms, each associated with an unknown reward distribution, with the goal of identifying the arm with the highest expected reward under a limited sampling budget. Best Arm Identification (BAI) formalizes this objective, most commonly in the fixed-budget setting, where the total number of pulls is predetermined and the probability of misidentifying the best arm is minimized (Bubeck et al., 2009; Audibert & Bubeck, 2010).

In many real-world applications, however, the sampling budget is neither fixed nor known in advance. Examples include clinical trials with fluctuating enrollment, large-scale crowdsourcing platforms, and online recommendation systems with dynamically expanding candidate pools. This motivates the study of *anytime* BAI algorithms, which maintain a valid estimate of the best arm at any stopping time without prior knowledge of the budget. As argued by Audibert & Bubeck (2010), the anytime formulation is a natural and practically relevant extension of BAI.

A fundamental challenge arises when the available budget is severely limited. In particular, when the total budget $T$ is smaller than the number of arms $N$, the learner enters a qualitatively different regime: some arms may never be sampled, and failure to identify the best arm may occur not because of estimation error, but because promising arms are never given the opportunity to compete. In this regime, the dominant difficulty lies in the algorithm's ability to *screen and retain sufficiently many viable candidates*, rather than in refining reward estimates.

Many existing anytime algorithms implicitly assume a minimum budget that allows each arm to be pulled at least once, which limits their effectiveness in such settings (Jourdan & Réda, 2023; Jun & Nowak, 2016; Degenne & Perchet, 2016). Several recent methods address this limitation by adapting the anytime framework to situations where resources are insufficient to sample all arms even once, often through bracketing or subsampling strategies (Katz-Samuels & Jamieson, 2020; Zhao et al., 2023). While these approaches provide theoretical guarantees on the misidentification probability under limited budgets, they typically define the "data-poor regime" solely in terms of the budget size $T$, without accounting for the algorithm-dependent ability to evaluate and retain promising arms.

To capture this distinction, we introduce the concept of the *data-poor condition*, which occurs when the available budget is insufficient for a given algorithm to meaningfully evaluate all potentially optimal arms. Crucially, this condition is algorithm-dependent: under the same budget constraint, different algorithms may retain vastly different sets of plausible candidates. The data-poor condition therefore delineates the regime in which the dominant source of error shifts from estimation noise to limited screening capacity and directly determines lower bounds on the probability of misidentifying the best arm. We formally define the data-poor condition and analyze its implications for anytime BAI in Section 4.1.

[1]Department of Statistics, Seoul National University, Seoul, South Korea. Correspondence to: Junyong Park <junyongpark@snu.ac.kr>.

*Proceedings of the 43rd International Conference on Machine Learning*, Seoul, South Korea. PMLR 306, 2026. Copyright 2026 by the author(s).

To address this challenge, we propose *Box Thirding* (B3), a novel anytime BAI algorithm tailored to operate effectively under the data-poor condition. B3 organizes arms into a hierarchical structure and performs repeated ternary comparisons, in which strong arms are promoted for further evaluation, weak arms are discarded, and uncertain arms are deferred for future reconsideration. This suspension mechanism prevents premature elimination while continuously allocating more budget to promising candidates. Importantly, B3 is fully anytime, requires no prior knowledge of the budget $T$, and introduces no tuning parameters. By design, B3 encounters the data-poor condition only when $T \lesssim N$, which is unavoidable since at least $T = N$ pulls are required to sample each arm once. The B3 algorithm is described in detail in Section 3.

Our theoretical analysis is structured around a decomposition of the overall error probability into two components: a non-inclusion probability, corresponding to the event that the best arm is never retained as a candidate under limited screening capacity, and a misidentification probability, capturing estimation error conditional on inclusion. This decomposition allows us to explicitly quantify the role of screening capacity in anytime BAI under limited budgets. We show that B3 achieves optimal screening capacity up to constant factors and attains sharp upper bounds on the probability of returning a suboptimal arm. Theoretical guarantees and comparisons with existing anytime methods are provided in Section 4.2, and numerical experiments validating our results are presented in Section 5.

**Conflict of Interest Disclosure** The authors declare that there are no financial conflicts of interest related to this work.

## 2. Preliminaries

### 2.1. Setup and Notations

The problem involves $N$ arms and a total sampling budget of $T$ pulls. Each arm $i \in \{1, \ldots, N\}$ generates rewards independently from a 1-sub-Gaussian distribution with an unknown mean $\mu_i$. That is, if $X$ denotes the reward obtained from pulling arm $i$, the sub-Gaussian assumption ensures that the probability of large deviations from the mean decays at least as fast as $\exp(-t^2/2)$. The mean rewards satisfy the ordering $\mu_1 > \mu_2 \geq \ldots \geq \mu_N$, which is unknown to the player.

For any subset of arms $A \subset \{1, 2, \ldots, N\}$, we define $\mu_*(A)$ as the highest mean among the arms in $A$:

$$\mu_*(A) = \max_{i \in A} \mu_i.$$

The *estimated best arm*, denoted by $a_T$, is the arm selected as the candidate for the best arm after $T$ pulls. This selection is based on the algorithm and the observed average rewards.

Finding the *true best arm* ($\mu_1$) is challenging with an unknown budget $T$, as it may not be sufficient to evaluate all $N$ arms. Instead, we aim to identify an $\epsilon$-best arm, which provides a practical and achievable alternative. An $\epsilon$-best arm is an arm whose mean reward satisfies $\mu_1 - \mu_i < \epsilon$. This relaxed objective ensures that the mean of the estimated best arm is close to $\mu_1$, even when exhaustive evaluations of all arms are infeasible.

### 2.2. Related Work

#### 2.2.1. THEORETICAL RESULTS OF BAI

In the fixed-budget BAI problem, the goal is to minimize the probability of misidentifying the best arm under a limited sampling budget (Audibert & Bubeck, 2010). Unlike standard multi-armed bandit problems that focus on cumulative regret, BAI algorithms are typically evaluated using simple regret, defined as $\mathbb{E}(\mu_1 - \mu_{a_T})$.

Another common evaluation criterion is the $(\epsilon, \delta)$-sample complexity, which characterizes the budget required to identify an $\epsilon$-best arm with high probability (Even-Dar et al., 2002; Kaufmann et al., 2016; Mannor & Tsitsiklis, 2004). Recent work has shown that this notion can be adapted to the fixed-budget setting by directly controlling the misidentification probability $\mathbb{P}(\mu_1 - \mu_{a_T} > \epsilon)$ as a function of the budget $T$ (Zhao et al., 2023). Such bounds simultaneously characterize both simple regret and $(\epsilon, \delta)$-sample complexity.

Regarding fundamental limits, Carpentier & Locatelli (2016) established lower bounds on the misidentification probability for fixed-budget BAI. In particular, when reward distributions are unknown, any algorithm satisfies

$$\mathbb{P}(\mu_1 \neq \mu_{a_T}) \geq \exp\left(-c \frac{T}{\left(\max_{i>1} \frac{i}{(\mu_1 - \mu_i)^2}\right) \log_2 N}\right),$$
(1)

for some universal constant $c > 0$, highlighting the intrinsic difficulty of large-scale BAI problems. Subsequent work further shows that these lower bounds are not uniformly achievable across all problem instances, especially when the number of arms is large (Ariu et al., 2021).

#### 2.2.2. COMPARISON OF KEY BAI ALGORITHMS IN FIXED BUDGET SETTING

**Uniform Sampling (US)** is the simplest approach, pulling all arms equally and selecting the arm with the largest average reward. Despite its simplicity, Wang et al. (2024) showed that US is admissible in the sense that for any algorithm, there exists at least one reward distribution on which

US performs better.

The **UCB-E** algorithm (Audibert & Bubeck, 2010) selects the arm with the highest upper confidence bound, calculated using the estimated mean reward and a tuning parameter. Unlike US, UCB-E adaptively allocates more samples to promising arms. Empirically, it significantly reduces the probability of misidentification compared to US, particularly when $N \gg 1$.

**Sequential Halving (SH)** (Karnin et al., 2013) iteratively allocates the budget across elimination rounds, pulling each remaining arm equally and discarding the bottom half at each stage. With an appropriate budget schedule, SH achieves a misidentification probability for unknown reward distributions that is bounded as in (1).

### 2.2.3. STRATEGIES FOR ALGORITHMS UNDER ANYTIME SETTING/DATA-POOR REGIME

In the fixed-budget setting, the total sampling budget must be specified in advance, whereas anytime algorithms operate without prior knowledge of the budget and must produce a valid recommendation at any stopping time.

A standard approach for converting fixed-budget BAI algorithms into anytime procedures is *the doubling strategy*, which repeatedly runs a fixed-budget algorithm under exponentially increasing budget guesses (e.g., $T_0, 2T_0, 4T_0$), restarting the algorithm until the true budget is exhausted (Audibert & Bubeck, 2010; Stephens, 2023; Zhao et al., 2023).

*The bracketing strategy* extends doubling to data-poor regimes by applying it to subsets of arms with increasing sizes. Rather than operating on all $N$ arms at each phase, bracketing selects a subset, runs doubling on that subset, and repeats the process with larger subsets if additional budget remains. The final recommendation is chosen by comparing the empirical best arms obtained from each subset.

By progressively increasing the subset size, bracketing enables anytime operation even when $T < N$, as in BUCB (Katz-Samuels & Jamieson, 2020) and BSH (Zhao et al., 2023). However, since information from earlier subsets is not fully integrated, performance is limited under severe budget constraints.

## 3. The Box Thirding Algorithm

The B3 algorithm is motivated by a hierarchical selection idea originating from robust statistics. Figure 1 illustrates this idea of the remedian estimator (Rousseeuw & Bassett Jr, 1990), which partitions data into small blocks and summarizes them by local medians. These intermediate medians act as coarse selection devices; nevertheless, repeated hierarchical selection yields an estimator that converges in

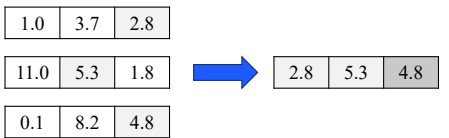

*Figure 1.* Toy example of remedian estimation: partition the data into three blocks and take their within-block medians (2.8, 5.3, 4.8); taking the median of these medians yields 4.8. Repeating this hierarchical "median-of-medians" construction produces an estimator that converges in probability to the population median.

probability to the true median.

In SH, a median-based screening mechanism appears: at each stage, the empirical median of rewards serves as a threshold, promoting arms above it and discarding those below. Although SH relies on the exact median of observed rewards, precise estimation is unnecessary due to reward noise and the corrective effect of subsequent rounds.

B3 adopts this selection philosophy in a local and hierarchical manner. By performing median-based elimination within boxes and combining the survivors across levels, it yields a consistent estimator of the effective median threshold while allowing early, noisy promotion errors to be corrected through further comparisons.

### 3.1. Box Operations and Main Algorithm

B3 algorithm processes arms iteratively by organizing them into hierarchical structures called *boxes*. Each box, denoted as $\text{Box}(l, j)$, stores up to three arms together with their average rewards, and serves as the basic unit for comparison and decision making.

The two parameters indicate the following:

- **Level** ($l$): Level represents the algorithm's current evidence of being optimal in an arm, where higher levels correspond to stronger candidates.
- **Deferment count** ($j$): Deferment count records the number of times the decision to promote or eliminate an arm has been postponed.

Whenever a box becomes full, B3 applies a ternary selection rule, implemented by the procedure ARRANGE_BOX. Given three arms in $\text{Box}(l, j)$, the procedure ranks them by their empirical means and assigns distinct roles with different sampling implications.

---

**Algorithm 1** ARRANGE_BOX

---

**Input:** level $l$, deferment count $j$, discard set $D$
LIFT(the largest arm in $\text{Box}(l, j)$, $l + 1$)
SHIFT(the median arm in $\text{Box}(l, j)$, $l, j + 1$)
DISCARD(the smallest arm in $\text{Box}(l, j)$, $D$)

---

The arm with the largest empirical mean is LIFTed: it receives an additional $\lceil r_0^l \rceil$ samples, its empirical mean is updated, and it is promoted to $\text{Box}(l+1, 0)$. The arm with the smallest empirical mean is DISCARDed and permanently removed from further consideration. The remaining arm is SHIFTed: it receives no additional samples and is moved to $\text{Box}(l, j+1)$, deferring the promotion or elimination decision to a later comparison.

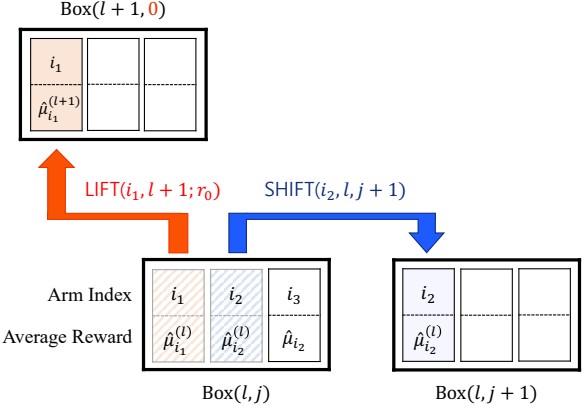

*Figure 2.* Illustration of ARRANGE_BOX$(l, j; D)$ when $\hat{\mu}_{i_1} > \hat{\mu}_{i_2} > \hat{\mu}_{i_3}$. The DISCARD operation is omitted for clarity.

The B3 algorithm, described in Algorithm 2, operates by repeatedly applying ARRANGE_BOX in two coordinated phases that prioritize promising arms while continuously introducing new candidates:

1. **Top-down evaluation sweep**: The algorithm scans boxes from the highest level downwards, ensuring that stronger candidates are prioritized for additional sampling and vacant higher-level boxes are promptly filled.
2. **Base-level replenishment**: After the sweep, if $\text{Box}(0, 0)$ has available capacity and unexamined arms remain, a new arm is introduced and lifted to the base level.

This iterative process enables B3 to continuously introduce new candidates while refining the estimates of promising arms, ensuring that a strong candidate for the best arm is maintained at any stopping time.

*Remark* 3.1. In Algorithm 2, the ranges indexed by $J_l$ are interpreted dynamically. That is, $J_l$ is updated during execution, and the inner loop proceeds according to the current state of the boxes at each iteration rather than a fixed range determined at initialization.

The choice of the base sampling parameter $r_0$ is guided by optimality considerations and is discussed in Proposition 4.11. When the budget is sufficient to examine all arms, B3 reduces to a modified procedure described in Appendix A.

---

**Algorithm 2** Box Thirding (B3)

**Initialize:** $L \leftarrow 0, J_L \leftarrow 0, t \leftarrow 0, D \leftarrow \emptyset$
Set $r_0$ which solves $r_0 + r_0^{1.5} = 4$
**while** The budget remains **do**
    **for** $l \in \{L, L-1, \dots, 0\}$ **do**
        **for** $j \in \{J_l, J_l - 1, \dots, 0\}$ **do**
            **if** $\text{Box}(l, j)$ is full and $\text{Box}(l+1, 0)$, $\text{Box}(l, j+1)$ are not full **then**
                ARRANGE_BOX$(l, j; D)$
            **end if**
            Update $J_l \leftarrow \max\{j : \text{Box}(l, j)$ is not empty$\}$
        **end for**
    **end for**
    Update $L \leftarrow \max\{l : \text{Box}(l, 0)$ is not empty$\}$
    Define $J_L \leftarrow \max\{j : \text{Box}(L, j)$ is not empty$\}$
    **if** $\text{Box}(0, 0)$ is not full **then**
        Select $i \notin D \sqcup (\bigsqcup_{l,j} \text{Box}(l, j))$ uniformly
        LIFT$(i, 0)$
    **end if**
**end while**
**Return:** The arm in $\text{Box}(L, 0)$ with the largest average

---

### 3.2. Justification of the Box Thirding Algorithm

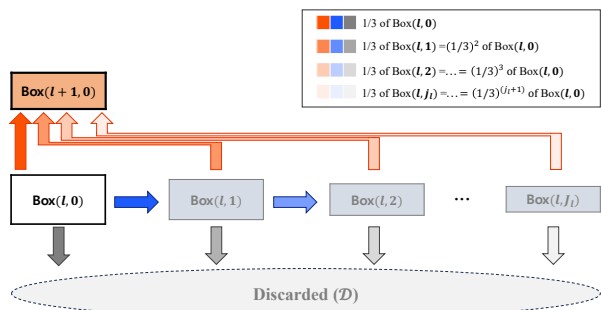

*Figure 3.* Fraction of arms that are lifted, shifted, and discarded at a fixed level $l$.

B3 implements an implicit halving procedure in an anytime manner. Despite relying on local ternary comparisons, its long-run screening behavior matches that of binary elimination schemes. Fix a level $l$ and consider the flow of arms originating from $\text{Box}(l, 0)$. At each application of ARRANGE_BOX, one arm is promoted to $\text{Box}(l+1, 0)$, while the remaining arms are either discarded or deferred. Applying the same ternary rule recursively to deferred arms, the fraction of arms eventually promoted from level $l$ to $l+1$ is

$$\frac{1}{3} + \frac{1}{3^2} + \frac{1}{3^3} + \cdots = \sum_{j=1}^{\infty} \frac{1}{3^j} = \frac{1}{2}.$$

Thus, although promotion decisions are made locally through ternary comparisons, B3 achieves an effective halving rate at the population level. The connection between the hierarchical ternary comparisons of B3 and the effective

halving behavior of SH will be made more explicit in the theoretical analysis of subsequent sections.

Crucially, this halving behavior is achieved without discarding past information. Deferred arms receive no additional samples, yet their empirical means are reused in subsequent comparisons against progressively refined sets of competitors. As weaker arms are filtered out, the median within $\text{Box}(l, j+1)$ becomes a more accurate decision threshold, allowing earlier noisy comparisons to contribute to increasingly reliable lift or discard decisions. This hierarchical reuse of past sampling results enables B3 to improve screening quality without additional sampling, which is essential for anytime operation under insufficient budgets.

## 4. Theoretical Analysis

This section provides theoretical guarantees for B3 under the data-poor condition. Throughout this section, asymptotic statements are taken with respect to the total sampling budget $T$, while allowing $N = N_T$ to scale with $T$. As discussed in Sections 3.2, when the sampling budget is insufficient to evaluate all arms, the primary challenge is whether the algorithm is able to retain the best arm as a viable candidate. To formalize this notion, we introduce the concept of a *candidate set*, which represents the collection of arms that remain under consideration at a given time.

Our analysis decomposes the overall error probability into two components: a *non-inclusion probability*, corresponding to the event that the best arm is never retained in the candidate set, and a *misidentification probability*, corresponding to selecting a suboptimal arm among the retained candidates due to estimation error. This decomposition reflects the fundamental trade-off between screening and estimation under limited budgets and forms the basis of our theoretical comparison. We show that B3 maximizes screening capacity under the data-poor condition and achieves sharp upper bounds on the probability of misidentifying an $\varepsilon$-best arm.

### 4.1. Candidate Set and Data-poor Condition

To analyze the screening behavior of B3 under limited budgets, we formalize the set of arms that remain under active consideration during the algorithm.

**Definition 4.1** (Candidate Set $C$). Let $(i_1, i_2, \ldots, i_N)$ denote the sequence of arms pulled by algorithm $\pi$ under budget $T$. Given this sequence, the candidate set $C \equiv C^\pi(i_1, \ldots, i_N; T)$ is defined as the collection of arms that could be selected as the best arm with non-zero probability under some reward distribution, i.e.,

$$\bigcup_{\nu \in \{\text{reward distribution}\}} \{i_k : \mathbb{P}_\nu (\mu_{i_k} = \mu_{a_T} \mid i_1, \ldots, i_N) > 0\}.$$

We denote $c_0 = |C|$, the cardinality of the candidate set.

For example, consider running B3 with budget $T = 6$, and suppose the arms are pulled in the order $(1, 2, 3, 4, 5)$.

- At $T = 1$–$3$: arms $(1, 2, 3)$ are each pulled once and lifted to the base comparison box, $\text{Box}(0, 0)$.
- At $T = 4$–$5$: the three arms in $\text{Box}(0, 0)$ are compared. The arm with the largest empirical mean is lifted to $\text{Box}(1, 0)$ after two additional pulls, the arm with the median empirical mean is shifted to $\text{Box}(0, 1)$, and the arm with the smallest empirical mean is discarded. As a result, $\text{Box}(0, 0)$ becomes empty.
- At $T = 6$: arm $4$ is pulled once and lifted to $\text{Box}(0, 0)$.

This illustrates the random permutation–based screening mechanism of B3, which continuously introduces previously unexamined arms under limited budgets. If the budget is exhausted at this point, B3 returns the arm in $\text{Box}(1, 0)$, since it is the arm residing at the highest level. Arm $5$ is never pulled and therefore cannot be selected as the best arm. Arm $4$, although pulled once, does not advance to higher levels and thus cannot be returned under any reward distribution. In contrast, arms $1, 2$, and $3$ each reach a level at which they could be selected as the output for some realization of rewards. Hence, the candidate set in this example is $C = \{1, 2, 3\}$, and the candidate set size is $c_0 = 3$.

This example highlights two key properties of the candidate set $C$ and its cardinality $c_0$. While the specific composition of $C$ depends on the random order in which arms are pulled—for instance, yielding $C = \{3, 5, 4\}$ under the permutation $(3, 5, 4, 1, 2)$— its size $c_0 = |C|$ is a deterministic quantity determined solely by the algorithm and the budget $T$. Moreover, not every arm that is sampled necessarily becomes a candidate: some arms may consume part of the budget yet fail to accumulate sufficient evidence to advance to higher levels, and therefore can never be selected as the output under any reward distribution.

When the budget is so restricted that $c_0$ cannot cover all potentially optimal arms, the algorithm enters what we define as the **data-poor condition**.

**Definition 4.2** (Data-Poor Condition). The *data-poor condition for $\epsilon$* occurs when the size of the Candidate Set, $c_0$, is smaller than the number of arms that are not $\epsilon$-best. That is:

$$c_0 \leq N - N_\epsilon,$$

where $N_\epsilon$ is the number of $\epsilon$-best arms. When $\epsilon = 0$, we refer to cases where $c_0 < N$ as simply the data-poor condition.

Unlike data-poor regime, this definition of the data-poor condition depends on the specific algorithm, as each algorithm determines $c_0$ differently.

## 4.2. Main Result

Our theoretical framework addresses the randomness induced by limited budgets by explicitly separating failures due to insufficient screening from errors arising in estimation after screening. This separation allows us to characterize the fundamental trade-off between retaining promising arms and accurately identifying the best arm once it is retained.

**Theorem 4.3.** *Under the data-poor condition for $\epsilon$, there exist universal constants $c_1, c_2 > 0$ independent of $T, N$, and the reward gaps $\Delta_i$ such that the B3 algorithm simultaneously satisfies the following upper bounds:*

*(a) (**Worst-case bound**)*

$$\mathbb{P}(\mu_1 - \mu_{a_T} > \epsilon) \leq \exp\left(-c_1 T \cdot \min\left\{\frac{N_{\epsilon/2}}{N}, \frac{\epsilon^2}{N}\right\}\right),$$

*where $N_{\epsilon/2}$ denotes the number of $\epsilon/2$-best arms.*

*(b) (**Instance-dependent bound**)*

$$\mathbb{P}(\mu_1 - \mu_{a_T} > \epsilon) \leq \log_2 N \cdot \exp\left(-c_2 \frac{T}{\log_2 N \cdot H_2(\epsilon)}\right),$$

*where the instance complexity $H_2(\epsilon)$ is defined as:*

$$H_2(\epsilon) := \frac{1}{N_{\epsilon/2}} \max_{i:\Delta_i \geq \epsilon} \frac{i}{\Delta_i^2}.$$

The worst-case bound in Theorem 4.3(a) directly follows from the observation that the overall failure event $\mu_1 - \mu_{a_T} > \epsilon$ is strictly contained in the union of two distinct error events:

$$\underbrace{(\mu_1 - \mu_*(C) > \epsilon/2)}_{\text{Non-Inclusion}} \cup \underbrace{(\mu_*(C) - \mu_{a_T} > \epsilon/2)}_{\text{Misidentification}}.$$

The non-inclusion term corresponds to the event that the true best arm is never retained in the candidate set $C$, which typically occurs under severe data-poor conditions when the arm is either sampled too late or omitted entirely during initial screening. In contrast, the misidentification term captures the standard estimation error that arises among the successfully screened arms within the candidate set.

While the worst-case analysis provides a robust framework to compare the overall bounds of B3 against US and BSH across the entire section, this structural decomposition allows us to go a step further. Specifically, by precisely tracking the behavior of individual arm gaps within these two error components, we establish the instance-dependent bound for B3 in Theorem 4.3(b). This refined rate captures the exact screening and identification efficiency of B3 in terms of the gap-dependent complexity $H_2(\epsilon)$, with its detailed formal proof deferred to Appendix D.5.

To prove the Theorem 4.3, we will show that

- $\mathbb{P}(\text{Non-inclusion}) \leq \exp\left\{-const \cdot \left(TN_{\epsilon/2}/N\right)\right\}$ in Section 4.3, and
- $\mathbb{P}(\text{Misidentification}) \leq \exp\left\{-const \cdot \left(T\epsilon^2/N\right)\right\}$ in Section 4.4.

If the main theorem holds, the following corollaries follow directly:

**Corollary 4.4** (Simple Regret). *Suppose there exists a constant $c_1 > 0$ such that the number of $\epsilon$-best arms satisfies $N_\epsilon \leq c_1 N\epsilon^{1/\alpha}$ for any $\epsilon > 0$.*

*Then, the simple regret of B3 satisfies:*

$$\mathbb{E}[\mu_1 - \mu_{a_T}] \leq C \max\left\{\frac{1}{T^\alpha}, \sqrt{\frac{N}{T}}\right\},$$

*where $C > 0$ is an absolute constant independent of $T$.*

**Corollary 4.5** (($\epsilon, \delta$)-Sample Complexity). *Let $o(N_{\epsilon/2})$ be the order of $N_{\epsilon/2}$ with respect to $\epsilon$. The minimum budget required to obtain a $\mathbb{P}(\mu_1 - \mu_{a_T} \leq \epsilon) \geq 1 - \delta$ guarantee is:*

$$T \geq const \cdot \left(\max\left\{\frac{N}{N_{\epsilon/2}}, \frac{N}{\epsilon^2}\right\} \ln\left(\frac{1}{\delta}\right)\right).$$

The proofs of Corollary 4.4 and Corollary 4.5 are provided in Appendix D.1.1 and Appendix D.1.2, respectively.

*Remark* 4.6. By combining the results of Corollary 4.9 and Proposition 4.12, we obtain the following explicit upper bounds on the $\epsilon$-best arm misidentification probability for US, BSH, and B3. Under the considered asymptotics, there exist absolute positive constants $(c_1, c_2, c_3, c_4)$ independent of $T$ and $N$ such that:

- US: $\exp\left(-c_1 \frac{TN_{\epsilon/2}}{N}\right) + N\exp\left(-c_2 \frac{T\epsilon^2}{N}\right)$
- BSH: $\exp\left(-c_3 \max(N_{\epsilon/2}, \epsilon^2) \cdot \frac{T}{N(\ln T)^2}\right)$
- B3: $\exp\left(-c_4 \max(N_{\epsilon/2}, \epsilon^2) \cdot \frac{T}{N}\right)$

These bounds highlight that B3 removes the sub-optimal $(\ln T)^2$ factor present in BSH, achieving a screening efficiency comparable to the data-poor regime's theoretical limits.

## 4.3. Non-Inclusion Probability of the $\epsilon/2$-Best Arm

The non-inclusion probability of the $\epsilon/2$-best arm arises from the *data-poor condition*. The following theorem provides an upper bound for this probability in terms of $c_0 = |C|$:

**Theorem 4.7.** *Let $N_\epsilon$ be the number of $\epsilon$-good arms, $C^\pi$ be the candidate set of an algorithm $\pi$, and $c_0^\pi$ be its cardinality. Under the data-poor condition for $\epsilon$, there exists an absolute*

*positive constant $c > 0$ independent of $N$, $c_0^\pi$, and $N_\epsilon$ such that the non-inclusion probability of the $\epsilon$-best arm is bounded as follows:*

$$\mathbb{P}(\mu_1 - \mu_*(C^\pi) > \epsilon) \leq \exp\left(-c\frac{c_0^\pi N_\epsilon}{N}\right).$$

The proof is provided in Appendix D.2.1.

We analyze the candidate set size $c_0$ under the data-poor condition for various algorithms:

**Proposition 4.8.** *Under the data-poor conditions, the sizes of the candidate sets ($c_0$) for different algorithms satisfy the following order-equivalence:*

- *US: $c_0 \asymp T$*
- *BSH: $c_0 \asymp \frac{T}{(\log_2 T)^2}$*
- *B3: $c_0 \asymp T$*

*where $A \asymp B$ denotes that $A$ and $B$ are bounded by each other up to absolute positive constants independent of $N$ and $\epsilon$.*

We deferred the proof of Proposition 4.8 to Appendix D.2.2.

Using Proposition 4.8 and Theorem 4.7, we can derive the order of the non-inclusion probabilities of $\epsilon/2$-best arm for different algorithms:

**Corollary 4.9.** *Under the data-poor condition for $\epsilon/2$, there exist absolute positive constants $c_1, c_2, c_3 > 0$ independent of $T, N$, and $\epsilon$ such that the non-inclusion probabilities of the $\epsilon/2$-best arm satisfy the following upper bounds:*

- *US: $\exp\left(-c_1\dfrac{TN_{\epsilon/2}}{N}\right)$*
- *BSH: $\exp\left(-c_2\dfrac{TN_{\epsilon/2}}{N(\ln T)^2}\right)$*
- *B3: $\exp\left(-c_3\dfrac{TN_{\epsilon/2}}{N}\right)$*

This follows by combining the candidate set sizes in Proposition 4.8 with the general non-inclusion bound in Theorem 4.7.

### 4.4. Misidentification Probability of the $\epsilon/2$-Best Arm Within Set $C$

#### 4.4.1. WORST-CASE UPPER BOUND OF SH

Identifying the best arm within the Candidate Set $C$ is equivalent to solving the BAI problem in a non-data-poor condition, where $C$ represents the entire set of arms. To analyze the $\epsilon/2$-best arm misidentification probability for B3, we first revisit SH, as it provides critical insights into the upper bounds for misidentification probabilities within the set $C$ of B3.

SH proceeds by iteratively allocating sampling budgets across elimination levels. At each level, all surviving arms are sampled equally, and approximately half of them are discarded based on their empirical means. The parameter $r_0$ controls the growth rate of the per-level budget $T_l$, which in turn determines how much sampling is concentrated at early versus later levels. In particular, larger values of $r_0$ spread the budget more evenly across levels, reducing the number of samples allocated at the base level.

The following theorem provides a bound on the misidentification probability for SH with sufficient budget $T$:

**Theorem 4.10.** *Consider the SH algorithm with total budget $T$, where the per-level budget is allocated as $T_l = \lceil r_0^l \rceil$ for some $r_0 \in (1, 2]$. Then there exist absolute positive constants $C_1, C_2 > 0$ independent of $T, N$, and $\epsilon$ such that the probability of misidentifying an $\epsilon$-best arm satisfies*

$$\mathbb{P}(\mu_1 - \mu_{a_T} > \epsilon) \leq \begin{cases} \exp\left(-C_1\dfrac{T\epsilon^2}{N}\right), & r_0 \in (1, 2), \\ \exp\left(-C_2\dfrac{T\epsilon^2}{N\log_2 N}\right), & r_0 = 2, \end{cases}$$

*reflecting a qualitative change in performance when the budget growth rate reaches the critical value $r_0 = 2$.*

Since the misidentification probability depends exponentially on the base pull count $t_0$, this abrupt change in the scaling of $t_0$ directly translates into the observed degradation in the error exponent at $r_0 = 2$.

The extended version of Theorem 4.10 along with its proof is provided in Appendix D.3.1.

Note that our result leads to a fundamentally tighter simple regret bound in the worst-case scenario compared to the existing guarantees in Zhao et al. (2023). Specifically, for the standard setting ($r_0 = 2$), the conventional misidentification probability bound is formulated as:

$$\mathbb{P}(\mu_1 - \mu_{a_T} > \epsilon) \leq \log_2 N \cdot \exp\left(-C\dfrac{T\epsilon^2}{N\log_2 N}\right)$$

where $C > 0$ is an absolute constant. Due to the presence of the $\log_2 N$ pre-factor, integrating this tail probability to derive the simple regret introduces an sub-optimal logarithmic overhead, failing to recover the information-theoretic minimax lower bound of $\Omega\left(\sqrt{N/T}\right)$. Furthermore, this probability bound becomes trivial (exceeds 1) as $N \to \infty$ unless the budget is large enough to satisfy $T = \Omega(N\log_2^2 N)$.

In contrast, by directly translating Theorem 4.10 into a simple regret framework, our bound remains non-trivial even under a strictly linear budget scaling $T = \mathcal{O}(N)$. Crucially, when $r_0 \in (1, 2)$, the resulting simple regret rate eliminates the sub-optimal logarithmic factors entirely and successfully matches the minimax lower bound $\Theta\left(\sqrt{N/T}\right)$ up

to absolute constants, establishing a tight optimality that matches the fundamental limits of simple regret (Lattimore & Szepesvári, 2020).

The budget allocation in SH can further minimize the upper bound, as shown below:

**Proposition 4.11.** *The upper bound of SH described in Theorem 4.10 with budget allocation $T_l = r_0^l$ in Algorithm 5 is minimized when $r_0$ solves $r_0 + r_0^{1.5} = 4$ ($r_0 \approx 1.728$).*

For the proof of Proposition 4.11, see Appendix D.3.2.

As explained in Section 3.2, B3 extends SH, and we will demonstrate in Proposition 4.12 that B3 achieves the same upper bound for the misidentification probability within the candidate set. Consequently, B3 allocates the budget for each level $l$ as $r_0^l$, where $r_0 \approx 1.728$.

### 4.4.2. Upper bounds for Anytime BAI

Building on the insights from SH, we extend the analysis to BSH and B3. The misidentification probability of $\epsilon/2$-best arm within set $C$ is bounded as follow:

**Proposition 4.12.** *Under the data-poor condition, potentially with $N = N_T$ varying with $T$, there exist absolute positive constants $c_1, c_2, c_3 > 0$ independent of $T, N$, and $\epsilon$ such that the probability of misidentifying the $\epsilon/2$-best arm within the candidate set $C$, $\mathbb{P}(\mu_*(C) - \mu_{a_T} > \epsilon/2)$, is bounded as follows:*

- *US:* $\leq N \exp\left(-c_1 \dfrac{T\epsilon^2}{N}\right)$
- *BSH:* $\leq \exp\left(-c_2 \dfrac{T\epsilon^2}{N(\ln T)^2}\right)$
- *B3:* $\leq \exp\left(-c_3 \dfrac{T\epsilon^2}{N}\right)$

The proof of Proposition 4.12 is shown in Appendix D.4.

## 5. Experiments

### 5.1. Setup

We evaluate the proposed B3 algorithm on the **New York Cartoon Caption Contest (NYCCC)** dataset, a standard benchmark for BAI (Jain et al., 2020; Yang et al., 2017; Mason et al., 2020; Katz-Samuels & Jamieson, 2020; Zhao et al., 2023). We focus exclusively on contest **893**, which contains $N = 5{,}513$ captions. Each caption receives categorical responses (*Not Funny*, *Somewhat Funny*, *Funny*). Following common practice, we define the true mean of each arm as the empirical proportion of *Funny* and *Somewhat Funny* responses after preprocessing identical to prior work. The resulting arm means exhibit a maximum–minimum range of approximately 0.529.

To disentangle the roles of *non-inclusion probability* and

*misidentification probability within the candidate set*, we generate rewards under three controlled noise regimes:

- **High-noise**: reward of arm $i \sim \mathcal{N}(\mu_i, 0.5^2)$,
- **Moderate-noise**: reward of arm $i \sim \mathcal{N}(\mu_i, 0.2^2)$,
- **Deterministic**: reward of arm $i = \mu_i$.

The deterministic setting represents an idealized regime in which, once the best arm is included in the candidate set, the misidentification probability within the set is exactly zero. In contrast, the high-noise setting induces substantial uncertainty even after inclusion, making the identification of the best arm within the candidate set highly nontrivial. These contrasting regimes highlight the importance of balancing screening capacity and within-set discrimination, which is the central motivation behind our theoretical error decomposition.

To complement the regret-based evaluation, we additionally report empirical candidate set sizes for the compared anytime algorithms in Appendix B.1. These experiments provide direct empirical evidence for the screening-capacity interpretation underlying the proposed non-inclusion analysis.

We compare B3 with US, BUCB ($\delta = 0.1$), and BSH under a fixed budget of $T = 10{,}000$. Since our primary focus is the anytime setting under insufficient sampling budgets, we restrict comparisons to algorithms that can operate without prior knowledge of the total budget $T$, including regimes where $T < N$. Accordingly, all compared methods are anytime algorithms capable of producing a valid recommendation at arbitrary stopping times. All results are averaged over 1,000 independent repetitions.

### 5.2. Results

Figure 4 illustrates how the relative contributions of non-inclusion probability and misidentification probability within the candidate set vary across noise regimes in the NYCCC 893 dataset. In the deterministic setting, simple regret depends solely on whether the best arm is included in the candidate set, leading US to achieve the lowest regret due to its maximal screening capacity, with B3 performing comparably.

As observation noise increases, however, misidentification within the candidate set becomes the dominant source of error, and methods that rely on broad but shallow screening deteriorate rapidly. Across all regimes, B3 consistently achieves strong performance by balancing candidate retention and within-set discrimination, demonstrating that the proposed decomposition of error is not only theoretically meaningful but also predictive of empirical performance under insufficient sampling budgets.

These results demonstrate two key points. First, B3 achieves

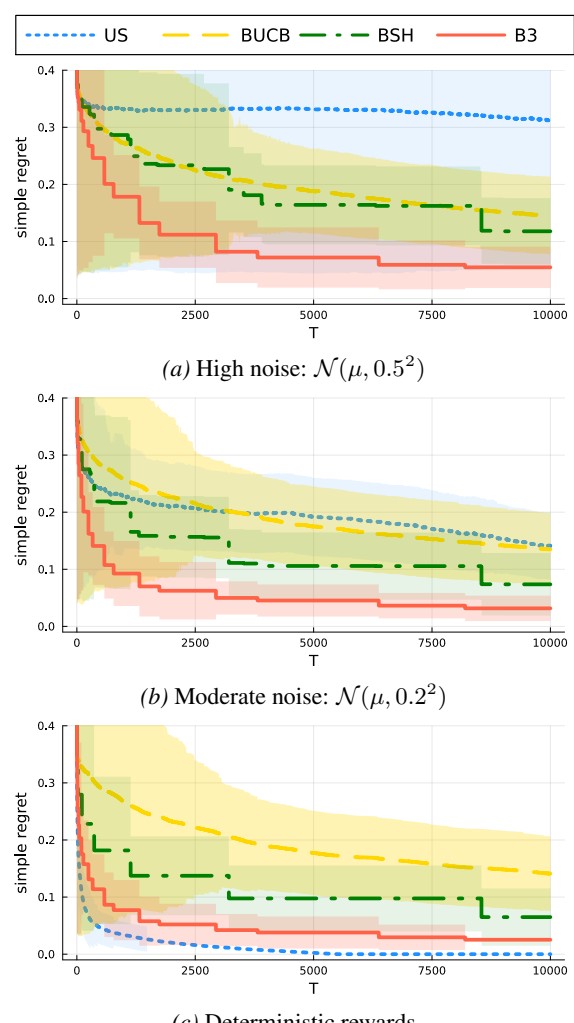

*(a) High noise:* $\mathcal{N}(\mu, 0.5^2)$

*(b) Moderate noise:* $\mathcal{N}(\mu, 0.2^2)$

*(c) Deterministic rewards*

*Figure 4.* Simulation results on the NYCCC 893 dataset under three reward noise regimes. Curves indicate the mean performance, and shaded regions correspond to the 25%–75% quantile range.

the most balanced trade-off between non-inclusion probability and misidentification probability within the candidate set, leading to robust performance across fundamentally different environments. Second, the empirical behavior across noise regimes aligns precisely with our theoretical error decomposition, confirming that separating the total error into non-inclusion and within-set misidentification is not merely a proof technique, but a practical tool for predicting the performance of anytime BAI algorithms under insufficient budget sampling.

Viewed through this lens, the error decomposition provides a simple and actionable guideline for choosing anytime BAI algorithms under limited budgets:

- **High reward variability:** prioritize algorithms with strong discrimination within the candidate set.
- **Low reward variability:** prioritize algorithms with large screening capacity (e.g., US or B3).

- **Unknown noise regime:** use B3 as a robust default, balancing both sources of error.

Additional experiments under alternative reward distributions (Bernoulli and power-law) as well as non–data-poor settings are reported in Appendix B.

## 6. Conclusion

In this work, we proposed the B3 algorithm for efficient operation under data-poor conditions. We formalized the notion of the data-poor regime and established simple regret and sample complexity bounds, demonstrating that B3 improves upon existing anytime algorithms in this setting.

Despite its efficiency, the current version of B3 discards samples from previous levels when arms are promoted or deferred. Recent work (e.g., Kone et al. (2024)) suggests that retaining and aggregating historical samples can substantially reduce estimation variance without weakening theoretical guarantees. Incorporating such sample reuse into B3's hierarchical box structure is a promising direction for future work and may further narrow the gap between anytime methods and fixed-budget algorithms, particularly in high-variance or complex environments.

## Code availability.

The implementation of Box Thirding is publicly available at `https://github.com/IAMSHHWANG/BoxThirding/`.

## Acknowledgements

This work was supported by the National Research Foundation of Korea (NRF) grant funded by the Korea government (MSIT) (RS-2025-00556575).

## Impact Statement

This work studies sample-efficient best-arm identification under limited budgets. Potential applications include adaptive experimentation and recommendation systems. As the contribution is primarily methodological, no immediate broader societal risks are identified beyond those associated with standard sequential decision-making systems.

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

## A. A Comprehensive Version of Box Thirding for Non-Data-Poor Conditions

Initially, the base level $l_B$ is set to 0, and the iteration level goes from $L$ to $l_B$. When all arms have been examined, $l_B$ is incremented by 1. Subsequently, all arms in the lowest-level boxes($\text{Box}(l_B - 1, \cdot)$) are lifted to higher levels. Additionally, arms previously discarded with $r_0^{l_B - 1}$ pulls are removed from the discarded set $D$ and reintroduced into the evaluation process, treating them as if they have not yet been fully examined.

---

**Algorithm 3** Box Thirding (B3)

---

  **Input:** Arms $N$, Budget $T$
  **Initialize:** $L \leftarrow 0, J_L \leftarrow 0, t \leftarrow 0, N_a \leftarrow 0, l_B \leftarrow 0, D \leftarrow \emptyset$
  **while** $t \leq T$ **do**
    **for** $l = L$ **to** $l_B$ and $j = J_l$ **to** 0 **do**
      **if** $\text{Box}(l, j)$ is full and $\text{Box}(l + 1, 0), \text{Box}(l, j + 1)$ are not full **then**
        ARRANGE_BOX$(l, j)$
      **end if**
    **end for**
    Update $L \leftarrow \max\{l : \text{Box}(l, 0) \text{ is not empty}\}$.
    $J_l \leftarrow \max\{j : \text{Box}(l, j) \text{ is not empty}\}$ for all $l \in \{l_B, \ldots, L\}$.
    **if** $\text{Box}(l_B, 0)$ is not full and $N_a < N$ **then**
      Select an arm $i \notin D \sqcup (\bigsqcup_{\forall l,j} \text{Box}(l, j))$
      LIFT$(i, l_B)$
      $N_a \leftarrow N_a + 1$
    **else if** $N_a = N$ **then**
      Call UPDATE_BASE_LEVEL$(l_B, N_a, t, D)$
    **end if**
  **end while**
  **Return:** The arm in $\text{Box}(L, 0)$ with the largest mean

---

---

**Algorithm 4** UPDATE_BASE_LEVEL

---

  **Input:** Level $l_B$, Arm Counter $N_a$, Total Pulls $t$, Discarded Set $D$
  $R \leftarrow \bigsqcup_{j \leq J_{l_B}} \text{Box}(F, j)$
  LIFT$(i, l_B + 1)$ for all arms $i \in R$
  $N_a \leftarrow N_a - |\{i \in D : \text{number of pulls} = r_0^{l_B}\}|$
  $D \leftarrow D \setminus \{i \in D : \text{number of pulls} = r_0^{l_B}\}$

---

## B. Further Numerical Results

### B.1. Visualization of the Candidate Set Sizes

To complement the regret-based evaluation in the main text, we compare the candidate set sizes $c_0$ of the considered anytime algorithms under varying sampling budgets. Recall that the candidate set size quantifies the number of arms that remain viable candidates for the final recommendation under some reward realization. As discussed in Section 4.1, $c_0$ directly characterizes the screening capacity of an algorithm and determines the non-inclusion probability under insufficient sampling budgets.

Figure 5 illustrates the growth of candidate set sizes as a function of the total budget $T$. Consistent with Proposition 4.8, both US and B3 exhibit candidate set sizes growing linearly with $T$, while BSH and BUCB retain substantially fewer candidates due to their bracketing-based budget allocation strategies.

The staircase structure observed for B3 reflects the hierarchical box mechanism of the algorithm. Since promotions occur only when boxes become full, the candidate set size increases in discrete jumps corresponding to the creation of new active comparison levels.

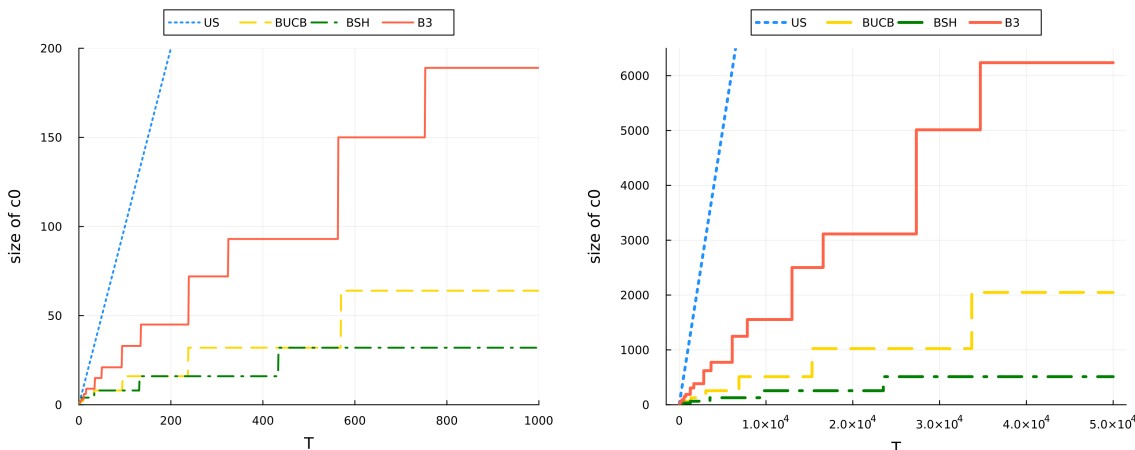

*Figure 5.* Empirical candidate set sizes $c_0$ as functions of the total budget $T$ for anytime algorithms under insufficient sampling budgets. Left: small-scale setting. Right: large-scale setting.

## B.2. Sensitivity Analysis on the choice of $r_0$

To evaluate the empirical robustness of the B3 algorithm, we conduct a sensitivity analysis on its primary structural hyperparameter, the budget growth rate $r_0 \in (1, 2]$. Under the exact same experimental configuration as the primary New Yorker Cartoon Caption Contest benchmark, we vary $r_0$ across a wide range of grid values, $\{1.3, 1.5, 2.0\}$, and compare their behaviors against our default choice of $r_0 \approx 1.7xx$ established in the theoretical framework.

As illustrated in Figure 6, the simple regret trajectories for all tested choices of $r_0$ exhibit a consistent trend, showing no substantial differences in performance across the entire spectrum of the sampling budget $T$. This empirical evidence demonstrates that B3 is exceptionally robust against the variations in its budget growth pace. Crucially, while we theoretically advocate for the choice of $r_0 \approx 1.7xx$ to satisfy the tightest analytical guarantees, our results indicate that practitioners can safely deploy the algorithm using standard benchmarks (such as $r_0 = 2.0$) or even smaller rates without suffering any noticeable performance degradation. Even under severe data-poor conditions ($T \leq N$), the candidate screening and selection capability remains highly stable. This parameter insensitivity underscores the practical utility of B3, as it effectively eliminates the need for tedious hyperparameter tuning in real-world pure exploration tasks.

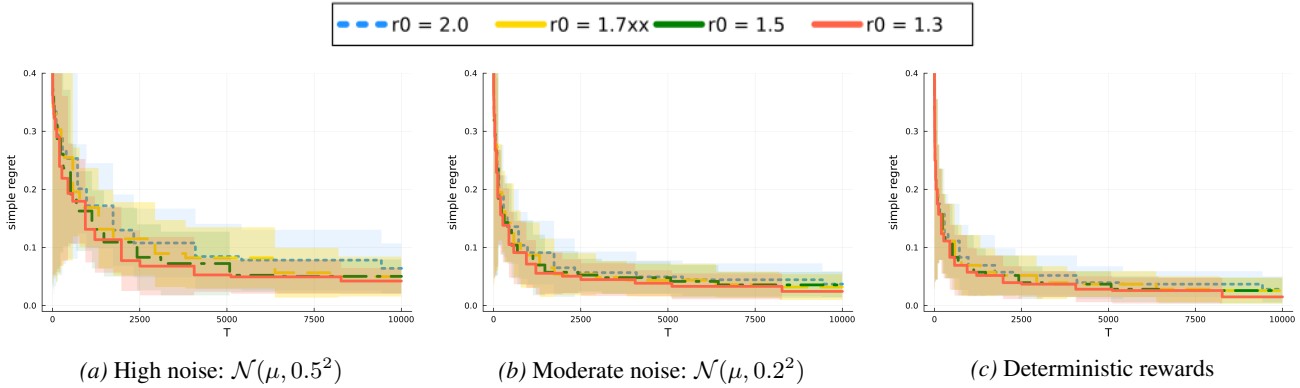

*(a)* High noise: $\mathcal{N}(\mu, 0.5^2)$      *(b)* Moderate noise: $\mathcal{N}(\mu, 0.2^2)$      *(c)* Deterministic rewards

*Figure 6.* Sensitivity analysis of the B3 algorithm under varying budget growth rates $r_0 \in \{1.3, 1.5, 2.0\}$ compared against our default configuration ($r_0 \approx 1.728$) on the New Yorker Cartoon Caption Contest dataset.

## B.3. Alternative Reward Distributions

To assess the robustness of Box Thirding beyond the Gaussian noise model used in the main text, we conduct additional experiments on the NYCCC 893 dataset under alternative reward distributions. Specifically, we replace the Gaussian reward noise with Bernoulli and power-law distributions, while keeping the underlying mean structure of the arms unchanged. This

allows us to isolate the effect of the reward distributional shape on algorithmic performance without altering the relative difficulty of the instance.

**Bernoulli rewards.** In the Bernoulli setting, rewards are generated as $X_{i,t} \sim \text{Bernoulli}(\mu_i)$, where $\mu_i$ denotes the true mean of arm $i$.

**Power-law rewards.** To model heavy-tailed noise, we generate rewards using a power-law–type construction based on the Kumaraswamy distribution. Specifically, each reward is obtained as

$$X_{i,s} \sim \text{Kumaraswamy}\left(\frac{1}{1/\tau_i - 1}, 1\right),$$

where $n_i(t)$ denotes the number of pulls of arm $i$ up to time $t$, and $\tau_i \in (0, 1)$ controls the tail heaviness. This construction yields a skewed, heavy-tailed reward distribution with finite mean but substantially larger variance than the Gaussian or Bernoulli cases, increasing the difficulty of within-set discrimination.

In the power-law setting, rewards are generated using a Kumaraswamy-based construction. Specifically, each pull yields a sample $X_{i,t} \sim \text{Kumaraswamy}\left(\frac{\mu_i}{1-\mu_i}, 1\right)$, which satisfies $\mathbb{E}[X_{i,t}] = \mu_i$ and $\text{Var}(X) = \mu_i(1 - \mu_i)^2/(2 - \mu_i)$. This construction yields a skewed, heavy-tailed reward distribution with finite mean but substantially larger variance than the Gaussian or Bernoulli cases, increasing the difficulty of within-set discrimination.

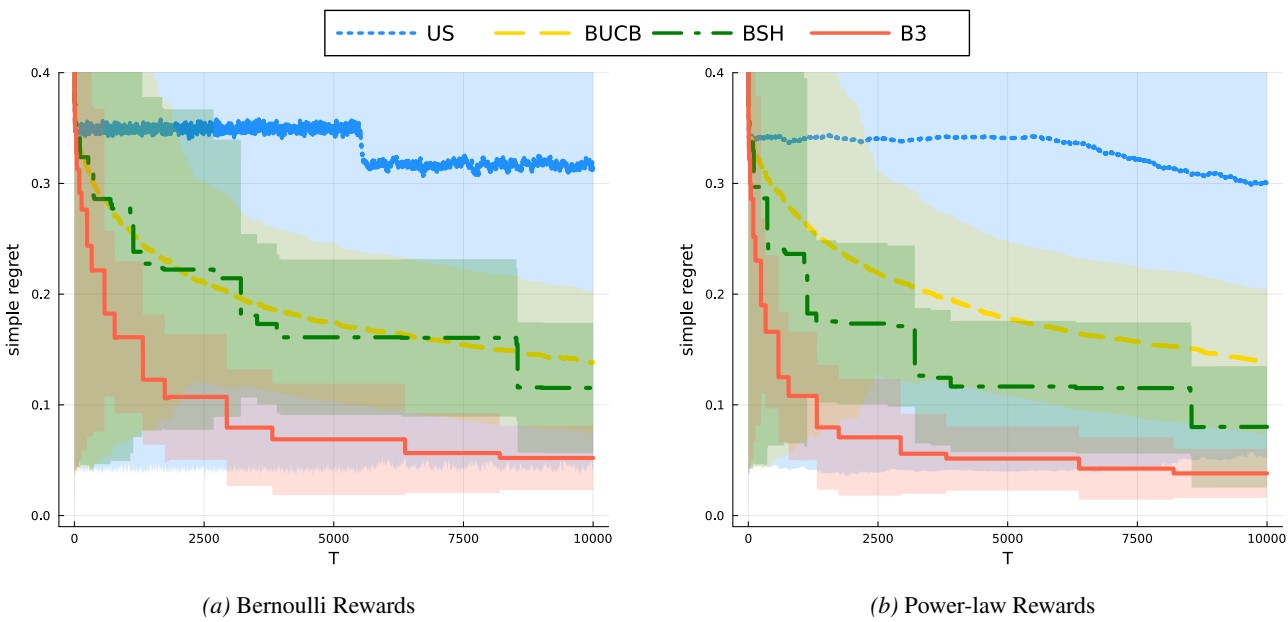

| | | | |
|---|---|---|---|
| (a) Bernoulli Rewards | | (b) Power-law Rewards | |

*Figure 7.* Simulation results on the NYCCC 893 dataset under different reward distributions. Curves indicate the mean performance, and shaded regions correspond to the 25%–75% quantile range.

**Results and discussion.** Across both alternative reward models, we observe consistent behavior with the Gaussian experiments reported in Section 5. In the Bernoulli setting, B3 maintains a balanced trade-off between candidate set inclusion and within-set misidentification, leading to stable simple regret across the sampling horizon. In the power-law setting, US exhibits little reduction in simple regret even as the budget $T$ increases. By contrast, BUCB and BSH achieve smaller simple regret than in the Bernoulli case, reflecting improved performance under this noise model. Overall, B3 consistently attains the smallest simple regret across budgets, indicating robust performance despite the increased difficulty of discrimination. Compared to Bernoulli rewards, the power-law distribution makes discrimination more challenging, suggesting that performance is driven less by non-inclusion probability and more by misidentification within the candidate set.

## B.4. Performance in Data-rich Regime $(c_0 = N)$

In this section, we provide additional experimental results to evaluate the robustness of the B3 algorithm in "data-rich" scenarios. These are settings where the budget $T$ is sufficiently large to allow all arms to be sampled at least once, typically satisfying $T \geq N \log_2 N$.

To assess the competitiveness of B3, we compare it against both anytime and fixed-budget baselines. It is important to note an information asymmetry in this setup: fixed-budget algorithms (SH and UCB-E) are provided with the exact value of the total budget $T$ as an input, whereas anytime algorithms (B3, BSH, and BUCB) operate without any prior knowledge of $T$.

The tables below summarize the performance of the algorithms in terms of simple regret. We report the average simple regret alongside the 25% and 75% quantiles to illustrate the performance distribution.

*Table 1.* Simple regret for $N = 128$ and $T = 896$ ($T = N \log_2 N$).

| Algorithm | Knowledge of $T$ | Average Simple Regret | 25% Quantile | 75% Quantile |
|---|---|---|---|---|
| B3 (Ours) | No (Anytime) | 0.0490 | 0.0290 | 0.0682 |
| SH ($r_0 = 1.7$) | Yes (Fixed) | 0.0446 | 0.0281 | 0.0652 |
| SH ($r_0 = 2.0$) | Yes (Fixed) | 0.0516 | 0.0290 | 0.0682 |
| UCB-E | Yes (Fixed) | 0.0664 | 0.0502 | 0.0749 |
| BSH | No (Anytime) | 0.0992 | 0.0587 | 0.1569 |
| BUCB | No (Anytime) | 0.1252 | 0.0682 | 0.1769 |

*Table 2.* Simple regret for $N = 1,024$ and $T = 10,240$ ($T = N \log_2 N$).

| Algorithm | Knowledge of $T$ | Average Simple Regret | 25% Quantile | 75% Quantile |
|---|---|---|---|---|
| B3 (Ours) | No (Anytime) | 0.0268 | 0.0000 | 0.0502 |
| SH ($r_0 = 1.7$) | Yes (Fixed) | 0.0195 | 0.0000 | 0.0290 |
| SH ($r_0 = 2.0$) | Yes (Fixed) | 0.0288 | 0.0000 | 0.0507 |
| UCB-E | Yes (Fixed) | 0.0403 | 0.0222 | 0.0588 |
| BSH | No (Anytime) | 0.1531 | 0.0661 | 0.2475 |
| BUCB | No (Anytime) | 0.2558 | 0.2374 | 0.3039 |

As shown in Tables 1 and 2, B3 maintains highly competitive performance even in the data-rich regime. Despite the disadvantage of not knowing the budget $T$, B3 achieves a simple regret comparable to that of algorithms with fixed budget and significantly outperforms existing anytime algorithms like BSH and BUCB. This suggests that the B3 mechanism is an efficient budget allocation strategy that generalizes well across different budget scales.

# C. Notations and Known Inequalities for Section D

## C.1. Notations

- $\lceil x \rceil$ : minimum integer which is larger than or equal to $x$
- $\lfloor x \rfloor$ : maximum integer which is smaller than or equal to $x$
- $C_l \equiv \{i : \text{arm } i \text{ has been in level } l\}$ with $c_l$ as its cardinality. This notation will be used for the proofs related to SH and B3.
- $A \sim B$ : asymptotic equivalence, i.e., $\lim_{B \to \infty} A/B = 1$.

## C.2. Some Equations & Inequalities

The followings are general inequalities that will be used for the proofs of the Theorems/Propositions/Corollaries in Section D.

1. **Sum of Power Series :** $\sum_{l=0}^{\infty}(l+1)r^l \leq (1-r)^{-2}, \forall r : 0 < r < 1$

2. **Bounding Summation by Integral :** $\sum_{l=c}^{L} f(l) \leq \int_c^L f(x)dx + f(c)$, when $f \geq 0$ is a continuous and non-increasing function.

3. **Chernoff Bound :** Suppose $X_i$ and $Y_i$ follow 1-subgaussian with mean $\mu_1$ and $\mu_2$ independently for $i \leq n$. If $\mu_X - \mu_Y = \Delta(> 0)$, then for the sample mean with $n$ samples $\hat{\mu}_X^n = (\sum_i X_i)/N, \hat{\mu}_Y^n = (\sum_i Y_i)/N$, $\mathbb{P}(\hat{\mu}_Y^n > \hat{\mu}_X^n) \leq \exp\{-\Delta^2 n\}$

4. **Gamma Parametrization :** $\int_0^\infty \exp(-x^k)dx = \frac{1}{k}\Gamma(\frac{1}{k})$, where $\Gamma(x) = \int_0^\infty \exp(-t)t^{x-1}dt$.

We attached useful Theorem that has been proved in Karnin et al. (2013); Zhao et al. (2023), and sketch of proof for the reader's convenience:

**Theorem C.1** (Karnin et al. (2013); Zhao et al. (2023)). *Let $N$ be the number of arms with associated means $\mu_1 \geq \mu_2 \geq \ldots \geq \mu_N$. For any $\epsilon > 0$, let $T_l$ be the number of times an arm is sampled.*

1. *The probability that the empirical average reward of an arm with a true mean less than $\mu_1 - \epsilon$ exceeds that of the arm with mean $\mu_1$ is bounded by:*
$$\mathbb{P}\left(\hat{\mu}_j^l > \hat{\mu}_1^l \mid \mu_j < \mu_1 - \epsilon\right) \leq \exp\left(-\epsilon^2 t_0 T_l\right).$$

2. *Furthermore, the probability that more than $N/2$ arms achieve this deviation is bounded by:*
$$\mathbb{P}\left(|\{i \neq 1 : \hat{\mu}_i^l > \hat{\mu}_1, \mu_1 - \epsilon > \mu_i\}| > N/2\right) \leq 3\exp\left(-t_0 T_l/8\right).$$

*Sketch of Proof.* The first upper bound follows directly from Chernoff's method.

For the second bound, consider the event:
$$\left\{\hat{\mu}_i^l > \hat{\mu}_1^l \text{ and } \mu_i < \mu_1 - \epsilon \text{ for more than } N/2 \text{ arms among the } N \text{ arms}\right\}.$$

This event is a subset of:
$$\left\{\hat{\mu}_1^l < \mu_1 - \frac{\epsilon}{2}\right\} \cup \left\{\text{ There are } \geq N/2 \text{ arms such that } \hat{\mu}_i^l > \mu_i - \frac{\epsilon}{2}\right\}.$$

We bound the probabilities of these two events separately:

1. $\mathbb{P}\left(\hat{\mu}_1^l < \mu_1 - \frac{\epsilon}{2}\right)$:

By Chernoff's bound, we have:
$$\mathbb{P}\left(\hat{\mu}_1^l < \mu_1 - \frac{\epsilon}{2}\right) \leq \exp\left(-\frac{\epsilon^2 t_0 T_l}{8}\right).$$

2. $\mathbb{P}\left(|\{i : \hat{\mu}_i^l > \mu_i + \frac{\epsilon}{2}\}| \geq \frac{N}{2}\right)$:

By Markov's inequality:
$$\mathbb{P}\left(\left|\{i : \hat{\mu}_i^l > \mu_i + \frac{\epsilon}{2}\}\right| \geq \frac{N}{2}\right) \leq \frac{\mathbb{E}\left|\{i : \hat{\mu}_i^l > \mu_i + \frac{\epsilon}{2}\}\right|}{N/2}.$$

Now, using the linearity of expectation and the fact that $\mathbb{P}(\hat{\mu}_i^l > \mu_i + \epsilon/2) \leq \exp\left(-\frac{\epsilon^2 t_0 T_l}{8}\right)$ for $i \neq 1$:

$$\frac{\sum_{i \neq 1} \mathbb{P}\left(\hat{\mu}_i^l > \mu_i + \frac{\epsilon}{2}\right)}{N/2} \leq \frac{N \cdot \exp\left(-\frac{\epsilon^2 t_0 T_l}{8}\right)}{N/2} = 2\exp\left(-\frac{\epsilon^2 t_0 T_l}{8}\right).$$

Combining these two bounds, we get:
$$\mathbb{P}\left(\text{more than } N/2 \text{ arms deviate as described}\right) \leq 3\exp\left(-\frac{\epsilon^2 t_0 T_l}{8}\right).$$

$\square$

# D. Proofs of Theoretical Results

The proof of Corollary 4.4 and Corollary 4.5 is straightforward from the result of Theorem **??**.

## D.1. Proofs of Section 4.2

### D.1.1. COROLLARY 4.4

In the following corollary, we adopt the $\alpha$-**parametrization** suggested by Jamieson et al. (2013), where the gap between the best arm and the $i$-th arm is defined as:

$$\mu_1 - \mu_i = \left(\frac{i-1}{N}\right)^\alpha.$$

This formulation ensures that the difficulty of identifying the best arm increases as $\alpha$ (where $\alpha > 0$) increases.

Under this setting, the number of $\epsilon$-best arms is given by:

$$N_\epsilon = \arg\max\{i : ((i-1)/N)^\alpha < \epsilon\},$$

which is equivalent to:

$$N_\epsilon = \arg\max\{i : (i-1) < N\epsilon^{1/\alpha}\}.$$

**Corollary D.1** (Simple Regret)**.** *Suppose that the number of $\epsilon$-best arms satisfies the approximation:*

$$N_\epsilon = \Theta(N \cdot \epsilon^{1/\alpha}).$$

*Then, the simple regret of B3 is bounded by:*

$$const \cdot \left(\max\left\{\frac{1}{T^\alpha}, \sqrt{\frac{N}{T}}\right\}\right).$$

*Proof.* By definition, the expected regret is given by:

$$\mathbb{E}[\mu_1 - \mu_{a_T}] = \int_0^\infty \mathbb{P}(\mu_1 - \mu_{a_T} > \epsilon)\, d\epsilon.$$

From **Theorem ??**, we have the probability bound:

$$\mathbb{P}(\mu_1 - \mu_{a_T} > \epsilon) \le \exp\left(-const \cdot \frac{T N_\epsilon}{N}\right) + \exp\left(-const \cdot \frac{T\epsilon^2}{N}\right).$$

Using the **Gamma parametrization** defined in Section C.2, namely:

$$\int_0^\infty \exp(-Kx^z)\, dx = \frac{1}{zK^{1/z}}\Gamma\left(\frac{1}{z}\right),$$

we obtain the following upper bounds:

$$\int_0^\infty \exp\left(-\Omega\left(\frac{T\epsilon^2}{N}\right)\right) d\epsilon = const \cdot \left(\sqrt{\frac{N}{T}}\right),$$

and

$$\int_0^\infty \exp\left(-T\epsilon^{1/\alpha}\right) d\epsilon = \mathcal{O}\left(\frac{1}{T^\alpha}\right).$$

$\square$

*Remark* D.2. A key observation is that the **critical value** of $\alpha$ is $1/2$. Specifically, when $\alpha \le 1/2$, the term $T^{-\alpha}$ is always smaller than $\sqrt{T/N}$, meaning that the dominant term in the regret bound depends on the problem difficulty parameter $\alpha$.

### D.1.2. COROLLARY 4.5

**Corollary D.3** (($\epsilon, \delta$)-Sample Complexity). *Let $o(N_{\epsilon/2})$ be the order of $N_{\epsilon/2}$ with respect to $\epsilon$. The minimum budget required to obtain a $\mathbb{P}(\mu_1 - \mu_{a_T} \le \epsilon) \ge 1 - \delta$ guarantee is:*

$$T \ge const \cdot \left(\max\left\{\frac{N}{N_{\epsilon/2}}, \frac{N}{\epsilon^2}\right\} \ln\left(\frac{1}{\delta}\right)\right).$$

*Proof.* By the result of Theorem **??**, we obtain the following probability bound:

$$\mathbb{P}(\mu_1 - \mu_{a_T} > \epsilon) \le \exp\left(-C_1 \cdot \left(\frac{TN_{\epsilon/2}}{N}\right)\right) + \exp\left(-C_2 \cdot \left(\frac{T\epsilon^2}{N}\right)\right)$$

for some constants $C_1, C_2 > 0$. To ensure that both terms on the right-hand side (RHS) are at most $\delta/2$, we impose the following conditions.

For the first term:

$$\exp\left(-const \cdot \left(\frac{TN_{\epsilon/2}}{N}\right)\right) \le \frac{\delta}{2} \implies T \ge const \cdot \frac{N}{N_{\epsilon/2}} \ln\left(\frac{2}{\delta}\right).$$

For the second term:

$$\exp\left(-const \cdot \left(\frac{T\epsilon^2}{N}\right)\right) \le \frac{\delta}{2} \implies T \ge const \cdot \frac{N}{\epsilon^2} \ln\left(\frac{2}{\delta}\right).$$

Combining these two conditions, we conclude:

$$T \ge const \cdot \left(\max\left\{\frac{N}{N_{\epsilon/2}}, \frac{N}{\epsilon^2}\right\} \ln\left(\frac{1}{\delta}\right)\right).$$

$\square$

### D.2. Proofs of Section 4.3

#### D.2.1. THEOREM 4.7

Although Theorem 4.7 specifically addresses the case under the data-poor condition, potentially with $N = N_T$ varying with $T$, we extend its result to cover all cases ($c_0 + N_\epsilon \ge N$) below.

**Theorem D.4** (Extended Statement of Theorem 4.7). *Let $N_\epsilon$ denote the number of $\epsilon$-good arms, $C$ the set of valid arms, and $c_0 = |C|$ its cardinality. The probability that an $\epsilon$-best arm is not included in $C$ is given as:*

$$\mathbb{P}(\mu_1 - \mu_1(C) > \epsilon) \le \begin{cases} 0, & \text{if } c_0 + N_\epsilon > N, \\ C_1 \cdot \left(\left(1 + \frac{N_\epsilon}{N - N_\epsilon}\right)^{-(N - N_\epsilon)} \cdot \left(\frac{N}{N_\epsilon}\right)^{-N_\epsilon}\right), & \text{if } c_0 + N_\epsilon = N, \\ \exp\left(-C_2\left(\frac{c_0 N_\epsilon}{N}\right)\right), & \text{if } c_0 + N_\epsilon < N. \end{cases}$$

*for some constants $C_1, C_2 > 0$.*

*Proof.* We provide a unified and intuitive proof for the non-inclusion probability across all regimes of the candidate set size $c_0$ and the number of $\epsilon$-best arms $N_\epsilon$.

The probability $P$ that no $\epsilon$-best arm is included in the candidate set $C$ is given by the hypergeometric ratio:

$$P(\mu_1 - \mu_1(C) > \epsilon) = \frac{\binom{N-N_\epsilon}{c_0}}{\binom{N}{c_0}} = \prod_{i=0}^{c_0-1}\left(1 - \frac{N_\epsilon}{N-i}\right) \tag{2}$$

**Case 1:** $c_0 + N_\epsilon > N$ (Sufficient Capacity)

In this regime, the number of candidate slots $c_0$ plus the number of $\epsilon$-best arms $N_\epsilon$ exceeds the total number of arms $N$. By the Pigeonhole Principle, it is impossible to select $c_0$ arms without picking at least one $\epsilon$-best arm. Mathematically, the product contains a term where $i = N - N_\epsilon$, making $(1 - \frac{N_\epsilon}{N-i}) = 0$. Thus:

$$P = 0 \tag{3}$$

**Case 2:** $c_0 + N_\epsilon = N$ (Critical Boundary) Here, the candidate set size is exactly equal to the number of arms that are not $\epsilon$-best. The only way to miss all $\epsilon$-best arms is to pick every single "bad" arm. The probability is:

$$P = \frac{1}{\binom{N}{N_\epsilon}} \leq \left(\frac{N_\epsilon}{N}\right)^{N_\epsilon} \tag{4}$$

Using the property $\binom{n}{k} \geq (n/k)^k$, we see that $P$ decays exponentially with the number of $\epsilon$-best arms, which is consistent with the $O(\cdot)$ bound in Theorem D.4.

**Case 3:** $c_0 + N_\epsilon < N$ (Data-Poor Condition)

To provide a more intuitive understanding of the non-inclusion probability, we present a simplified proof using the fundamental inequality $1 - x \leq e^{-x}$.

Under the data-poor condition where $c_0 + N_\epsilon < N$, the probability $P$ that none of the $N_\epsilon$ arms belonging to the $\epsilon$-best set are included in the candidate set $C$ of size $c_0$ can be expressed as a ratio of combinations:

$$P = \frac{\binom{N-N_\epsilon}{c_0}}{\binom{N}{c_0}} \tag{5}$$

By expanding the binomial coefficients, we can rewrite this probability as a product of $c_0$ terms:

$$P = \prod_{i=0}^{c_0-1} \frac{N - N_\epsilon - i}{N - i} = \prod_{i=0}^{c_0-1}\left(1 - \frac{N_\epsilon}{N-i}\right) \tag{6}$$

Applying the inequality $1 - x \leq \exp(-x)$ for each term in the product, we obtain:

$$P \leq \prod_{i=0}^{c_0-1} \exp\left(-\frac{N_\epsilon}{N-i}\right) = \exp\left(-\sum_{i=0}^{c_0-1} \frac{N_\epsilon}{N-i}\right). \tag{7}$$

Since $N - i \leq N$ for all $i \geq 0$, it follows that $\frac{1}{N-i} \geq \frac{1}{N}$. We can lower bound the summation in the exponent as:

$$\sum_{i=0}^{c_0-1} \frac{N_\epsilon}{N-i} \geq \sum_{i=0}^{c_0-1} \frac{N_\epsilon}{N} = \frac{c_0 N_\epsilon}{N} \tag{8}$$

Substituting this back into the exponential bound, we arrive at:

$$P \leq \exp\left(-\frac{c_0 N_\epsilon}{N}\right) \tag{9}$$

$\square$

By combining these cases, we observe a phase transition in the non-inclusion probability: 1. It is identically zero when the budget allows for sufficient screening ($c_0 > N - N_\epsilon$). 2. It is exponentially small at the boundary ($c_0 = N - N_\epsilon$). 3. It follows the rate $\exp(-\Omega(c_0 N_\epsilon / N))$ in the data-poor regime, where the dominant source of error is the limited screening capacity of the algorithm.

### D.2.2. PROPOSITION 4.8

**Proposition D.5.** *Under the data-poor condition, the sizes of the Candidate Sets($c_0$) for different algorithms are:*

- *US: $c_0 \asymp T$*
- *BSH: $c_0 \asymp T(\log_2 T)^{-2}$*
- *B3: $c_0 \asymp T$*

*Proof.* The proof for US is straightforward.

**Proof of BSH :** $c_0 \asymp T/(\ln T)^2$

The $B^{\text{th}}$ bracket is constructed when $t = (B-1)2^{B-1}$. After its creation, this bracket requires an additional budget of $B2^B \times B$ to return the best arm identified within bracket $B$. In the best case, all $B$ brackets contain non-overlapping arms. Therefore, the total number of valid arms after the $B^{\text{th}}$ bracket returns the best arm is

$$2^1 + 2^2 + \cdots + 2^B = 2^{B+1} - 2.$$

In the worst case, the arms in the first $B - 1$ brackets are subsets of the arms in bracket $B$. Consequently, the total number of valid arms is at least

$$2^B.$$

To generalize this, we define

$$\mathcal{B} = \max\{2^b : (b-1)2^{b-1} + b2^b \times b \le T\}.$$

Using this, the total number of valid arms is

$$c_0 = 2\mathcal{B} - 2.$$

To bound $\mathcal{B}$, consider first an upper bound. Simplifying the condition, we approximate

$$\mathcal{B} = \max\{2^b : (2b^2 + b - 1)2^b \le 2T\}.$$

Since $(2b^2 + b - 1)$ grows faster compared to $2b^2$ for all $b \in \{1, 2, \ldots\}$, $\mathcal{B}$ can be bounded by

$$\mathcal{B} \le \max\{2^b : b^2 2^b \le T\}.$$

Expanding this, we have

$$\mathcal{B} \le \frac{T}{\left(\log_2\left(\frac{T}{(\log_2(\frac{T}{\cdots}))^2}\right)\right)^2} \le \frac{T}{(\log_2(T) - \log_2\log_2(T))^2}.$$

For a lower bound on $\mathcal{B}$, we observe that

$$\mathcal{B} \ge \max\{2^b : (b+1)^2 2^{b+1} \le 4T\}.$$

This simplifies further to

$$\mathcal{B} \ge \frac{4T}{(\log_2(4T))^2}.$$

Combining these bounds, we derive the following bounds for $c_0 : \mathcal{B} \le c_0 \le 2\mathcal{B} - 1$:

$$\frac{4T}{(\log_2(4T))^2} \le c_0 \le \frac{2T}{(\log_2 T - \log_2\log_2 T)^2} - 1.$$

Both the lower bound and the upper bound for $c_0$ are of order $T/(\log_2 T)^2$. Therefore, we conclude that $c_0 = \Theta(T/(\log_2 T)^2)$.

**Proof of B3 :** $c_0 \asymp T$

To make the analysis rigorous, we explicitly define the analytical sets and notation. Let $N_{total}$ be the total number of arms that have ever been in level $l$. To distinguish between the candidate set $C$ and the arms that merely pass through a box, we define

$$C_{l,0} = \{i \in C \mid \text{arm } i \text{ experienced a full } Box(l,0)\}$$

and $n_l = |C_{l,0}|$. By definition, $C_{0,0} = C$ since all arms in the candidate set $C$ must, at some point, have experienced a full $Box(0,0)$ to be considered viable. Therefore, $c_0 = n_0$.

Then the number of arms $n_{l+1}$ that are promoted to level $l+1$ from level $l$ is given by:

$$n_{l+1} = \left\lfloor \frac{n_l}{3} \right\rfloor + \left\lfloor \frac{n_l}{3^2} \right\rfloor + \ldots + \left\lfloor \frac{n_l}{3^{\lfloor \log_3 n_l \rfloor}} \right\rfloor = \sum_{i=1}^{\lfloor \log_3 n_l \rfloor} \left\lfloor \frac{n_l}{3^i} \right\rfloor$$

Using $x - 1 \leq \lfloor x \rfloor \leq x$, we can bound $n_{l+1}$ as:

- $n_{l+1} \leq \frac{n_l}{3} \left( \frac{1 - 3^{-\log_3 n_l}}{1 - 3^{-1}} \right) = \frac{n_l}{3} \left( \frac{1 - 1/n_l}{2/3} \right) = \frac{n_l - 1}{2}$.
  Iteratively, we get $n_l \leq \frac{n_0 + 1}{2^L} - 1$.
- $n_{l+1} \geq \frac{n_l}{3} \left( \frac{1 - 3^{-\log_3 n_l + 1}}{1 - 3^{-1}} \right) - \log_3 n_l = \frac{n_l - 3}{2} - \log_3 n_l$.
  Since $n_l \leq 3n_{l+1}$, $n_{l+1}$ is larger than or equal to $\frac{n_l - 3}{2} - \log_3 3n_{l+1}$. With this lower bound of $n_l \geq \frac{n_{l-1} - 3}{2} - \log_3 n_{l-1} \geq \frac{n_{l-1} - 3}{2} - 1 - \log_3 n_l$ that matches $n_l$ on the LHS and RHS, we can get the iterative formulation that:

$$
\begin{aligned}
n_l &\geq \frac{\frac{n_{l-2} - 3}{2} - \log_3 n_{l-1} - 1 - 3}{2} - 1 - \log_3 n_l \\
&\geq \frac{\frac{n_{l-2} - 3}{2} - \log_3 n_l - 2 - 3}{2} - 1 - \log_3 n_l \\
&= \frac{n_{l-2}}{2^2} - (1/2^2 + 1/2^1) \cdot 3 - (2/2^1 + 1/2^0) - (1/2 + 1) \log_3 n_l \\
&\geq \ldots \geq \frac{n_0}{2^l} - 3 - 2 \sum_{j=1}^{\infty} j/2^j - 2 \log_3 n_l \\
&= \frac{n_0}{2^l} - 7 - 2 \log_3 n_l.
\end{aligned}
$$

If $n_L > 3$, then $n_{L+1} \geq 1$ so it is contradiction to the definition of $L = \arg\max\{l : Box(l,0) \text{ is not empty}\}$. i.e., $1 \leq n_L \leq 3$. Combining with the lower/upper bound of $n_L$ obtained above,

$$\frac{n_0}{2^L} - 7 - 2 \log_3 3 \leq n_L \leq \frac{n_0 + 1}{2^L} - 1.$$

By combining with $1 \leq n_L \leq 3$, we get two inequalities:

$$
\begin{aligned}
1 &\leq \frac{n_0 + 1}{2^L} - 1 \iff 2^{L+1} \leq n_0 + 1 \iff L \leq \log_2(n_0 + 1) - 1, \\
\frac{n_0}{2^L} - 9 &\leq 3 \iff \frac{n_0}{12} \leq 2^L \iff \log_2 n_0 - \log_2 12 \leq L.
\end{aligned}
$$

Fix $T_l = r^l$ with $1 < r < 2$. We define the theoretical budget $T_C$ spent on the candidate arms. The budget used in level $l$ for candidate arms is $T^l$, then $T^l$ equals to $n_l \times r^l$. With bounds of $n_l$, we get the bounds of the total budget $T_C = \sum_l n_l r^l$ for

the candidate set as:

$$T_C \le \sum_l \left( (n_0 + 1) \cdot \left(\frac{r}{2}\right)^l - r^l \right)$$

$$\le \left( (n_0 + 1)\frac{2}{2-r} - \frac{r^{L+1}-1}{r-1} \right)$$

$$\underset{(*)}{\le} \left( \frac{2(n_0+1)}{2-r} - \frac{(n_0+1)^{\log_2 r} - 1}{r-1} \right).$$

The last inequality in $(*)$ is derived from $L + 1 \le \log_2(n_0 + 1)$. Furthermore,

$$T_C \ge \sum_l \left(\frac{n_0}{2^l} - 9\right) r^l$$

$$\ge \left( \frac{2n_0}{2-r} - 9\frac{(n_0+1)^{\log_2 r} - 1}{r-1} \right).$$

Given the total budget $T$, we can infer the number of initial candidate arms $n_0$ that can be effectively processed. This implies that $T_C = \Theta(n_0)$. Since $n_0$ is the number of initial candidate arms, and the total budget $T$ must be sufficient to process these arms, it follows that $T_C = \Theta(T)$. Specifically, we can test at least

$$c^L = \max \left\{ n_0 : \left( \frac{2n_0}{2-r} - 9\frac{(n_0+1)^{\log_2 r} - 1}{r-1} \right) \le T \right\} \ge \frac{T(2-r)}{2n_0}$$

arms, and at most

$$c^U = \min \left\{ n_0 : T \le \left( \frac{2(n_0+1)}{2-r} - \frac{(n_0+1)^{\log_2 r} - 1}{r-1} \right) \right\}$$

arms.

Note that $\log_2 r < 1$ so that the leading term of both $c^U, c^L$ is $n_0$. i.e., $T \sim n_0$. Therefore, we can conclude that $c_0 = \Theta(T)$.

We showed that at least $c^L = const \cdot T$ arms had passed Stage 0. With $c^L$ arms, there are at least $\log_2 c^L - \log_2 12$ levels, and if the arm is tested before $c^s$ such that

$$2 \le const\frac{c^s + 1}{T}$$

i.e., the arm tested before the $const \cdot T$th has the chance to be selected as the best arm. $\qquad\square$

*Remark* D.6. Although B3 begins by pulling each of the first examined arms only once (i.e., Box$(0, 0)$), we can generalize this by starting with $t_0$ pulls in the initial examination. The budget allocated for pulling at level $l$ can then be scaled proportionally to $t_0 T_l$ pulls.

In this scenario, the total budget required to process $n_0$ initial arms through levels 0 to $L$, denoted as $T^{1:L}(t_0)$, increases linearly with $t_0$, i.e.,

$$T^{1:L}(t_0) = t_0 T^{1:L}(1).$$

Consequently, for a fixed total budget $T$, the number of arms $n_0$ that the algorithm can vet scales inversely with $t_0$. Specifically, we obtain the relationship $n_0 = \Theta(T/t_0)$, and the lower bound for the capacity $c^L$ (previously derived for $t_0 = 1$) is adjusted as:

$$c^L \ge \frac{T(2-r)}{2t_0}.$$

This confirms that even with a larger initial pull count $t_0$, the linear relationship between the sampling budget and the candidate screening capacity $c_0$ remains preserved, which is a key property of B3 in the data-poor regime.

*Remark* D.7. As described in Section 3.2, approximately half of the arms placed in level $l$ progress to level $l + 1$. If we apply SH with $T_l = r^l$ for some $r \in (1, 2)$, the maximum level generated is approximately $L \approx \log_2 N$, and the required budget scales as $T = \Theta(N)$. In other words, the budget $T$ allows for the complete examination of approximately $T$ arms in SH.

Our approach exhibits a similar order for the maximum level, satisfying $L = \Theta(\log_2 T)$, while also ensuring that the number of arms with a chance to be the best arm scales as $c_0 = \Theta(T)$, all without requiring prior knowledge of $T$.

## D.3. Proofs of Section 4.4

### D.3.1. THEOREM 4.10

Although Theorem 4.10 establishes an upper bound for SH under the specific choice $T_l = r_0^l$, the same argument extends to a broader class of sequences $\{T_l\}$. To show this, we proceed as follows:

1. We first analyze the cases where $T_l = r^l$ with $r_0 \in (1, 2]$, as stated in Theorem 4.10.

2. Next, we derive an upper bound for $T_l = (l+1)r^l$ for any $r \in (1, 2)$, following a similar approach as in the case $T_l = r^l$ (Lemma D.11).

3. Finally, we extend this technique to show that it applies to the broader range $\Omega(l^{4+\eta}) \le T_l \le const \cdot (2^l)$ (Lemma D.12).

Before presenting the extended version of Theorem 4.10, we introduce the following Lemma, which will be useful in our analysis.

**Lemma D.8** (Error function bound (Chiani et al., 2003)). *For any $a > 0$ and $z > 0$, the following upper bound holds:*

$$\int_z^\infty \exp(-x^2)dx \le \exp(-z^2).$$

Applying a change of variables, we obtain a more general bound.

**Lemma D.9** (Modified Lemma D.8). *For any $a > 0$ and $z > 0$, we have:*

$$\int_z^\infty \exp(-ax^2)dx \le \frac{1}{\sqrt{a}}\exp(-az^2).$$

*Proof.* Using the substitution $u = \sqrt{a}x$, which gives $du = \sqrt{a}dx$, we transform the integral as follows:

$$\int_z^\infty \exp(-ax^2)dx = \frac{1}{\sqrt{a}}\int_{\sqrt{a}z}^\infty \exp(-u^2)du.$$

Applying the bound from the previous lemma at $\sqrt{a}z$, we obtain:

$$\int_{\sqrt{a}z}^\infty \exp(-u^2)du \le \exp(-az^2).$$

Thus, we conclude:

$$\int_z^\infty \exp(-ax^2)dx \le \frac{1}{\sqrt{a}}\exp(-az^2).$$

$\square$

**Theorem D.10** (Theorem 4.10). *Consider the SH algorithm with a budget schedule $T_l = \lceil r_0^l \rceil$ for $r_0 \in (1, 2]$. The misidentification probability is bounded as follows:*

$$\mathbb{P}(\mu_1 - \mu_{a_T} > \epsilon) \le \begin{cases} \exp\left\{-C_1\left(\frac{T\epsilon^2}{N}\right)\right\}, & \text{if } r_0 \in (1, 2), \\ \exp\left\{-C_2\left(\frac{T\epsilon^2}{N\log_2 N}\right)\right\}, & \text{if } r_0 = 2, \end{cases}$$

*for some constants $C_1, C_2 > 0$.*

*Proof.* Before proving the Theorem 4.10, we formalize the SH algorithm as follow:

---

**Algorithm 5** Sequential Halving (SH, Karnin et al. (2013))

---

**Input:** $N$ (Number of arms), $T$ (Total budget), $T_l$ (Budget schedule per level)
Set $t_0 \leftarrow \max\{t : t \cdot \sum_{l=0}^{\lfloor \log_2 N \rfloor} T_l/2^l \leq T/N\}$
Initialize $C_0 \leftarrow [N]$ (Set of all arms)
**for** $l = 0, \ldots, \lfloor \log_2 N \rfloor$ **do**
    Pull each arm $i \in C_l$ for $t_0 \cdot T_l$ times
    $C_{l+1} \leftarrow \{\text{Top } \lceil |C_l|/2 \rceil \text{ arms in } C_l\}$
**end for**
**Return:** The arm in $C_{\lfloor \log_2 N \rfloor + 1}$

---

As described in Algorithm 5, we denote $t_0$ as the initial budget that satisfies

$$t_0 = \operatorname{argmax}\{t : tN(T_0 + T_1/2 + \ldots + T_L/2^L) \leq T\}.$$

Suppose we establish the following two properties:

1. For any sequence $(p_l)_{l=0}^{L}$ such that $\sum_{l=0}^{L} p_l \leq 1$ and $p_l \geq 0$, the event $\mu_1 - \mu_{a_T} > \epsilon$ is a subset of the union of events

$$\bigcup_{l=0}^{L} (\mu_1(C_l) - \mu_1(C_{l+1}) > p_l \epsilon).$$

2. The probability of each individual event satisfies the bound

$$\mathbb{P}(\mu_1(C_l) - \mu_1(C_{l+1}) > p_l \epsilon) \leq 3 \exp(-K\epsilon^2 t_0 (l+1)^2).$$

3. the initial budget $t_0 = const \cdot T/N$ when $r \in (0,1)$ and $t_0 = const \cdot T/(N \log_2 N)$ when $r = 2$.

Using Bounding Summation by Integral (as discussed in Section C.2) and applying Lemma D.8, we derive:

$$\mathbb{P}(\mu_1 - \mu_{a_T} > \epsilon) \leq \exp(-\Omega(K\epsilon^2 t_0)).$$

Applying Bounding Summation by Integral introduces the following inequality:

$$\sum_{l=0}^{L} \exp(-K\epsilon^2 t_0 (l+1)^2)$$
$$\leq \sum_{l=1}^{L} \exp(-K\epsilon^2 t_0 l^2)$$
$$\leq \int_1^{\infty} \exp(-K\epsilon^2 t_0 x^2) dx + \exp(-K\epsilon^2 t_0),$$

and Lemma D.8 bounds the integral part in above equation by:

$$\int_1^{\infty} \exp(-K\epsilon^2 t_0 x^2) dx \leq \frac{1}{\sqrt{K\epsilon^2 t_0}} \exp(-K\epsilon^2 t_0).$$

Thus,

$$\mathbb{P}(\mu_1 - \mu_{a_T} > \epsilon) \leq 3 \left( \frac{1}{\sqrt{K\epsilon^2 t_0}} + 1 \right) \cdot \exp(-K\epsilon^2 t_0).$$

Since the probability $\mathbb{P}(\mu_1 - \mu_{a_T} > \epsilon)$ is equivalent to $\mathbb{P}(\mu_1 - \mu_{a_T} \geq \Delta)$, where $\Delta$ is the smallest gap between $\mu_1$ and any $\mu_i$ that is strictly greater than zero, we can refine the bound. Specifically, if $\epsilon < \Delta$, we can replace $\epsilon$ in the probability expression with $\Delta$, yielding:

$$\mathbb{P}(\mu_1 - \mu_{a_T} > \epsilon) \leq \frac{1}{\sqrt{K(\epsilon \vee \Delta)^2 t_0}} \exp(-K\epsilon^2 t_0) \leq \exp\{-const \cdot (K\epsilon^2 t_0)\}.$$

As we substitute $t_0$ as $T/N$ or $T/(N \log_2 N)$ corresponding to $r$, we get the desired result.

To finalize the argument, it is necessary to rigorously establish the three key properties initially assumed.

### 1. Subset Inclusion

Let $(p_l)_{l=0}^L$ be a sequence satisfying $\sum_{l=0}^L p_l \leq 1$. Suppose that at each level, at least one $p_l\epsilon$-best arm is promoted to the next level, ensuring that the following condition holds:

$$\mu_1(C_l) - \mu_1(C_{l+1}) \leq p_l\epsilon.$$

Since the estimated best arm has a true mean of $\mu_1(C_{L+1})$, we can express the regret as:

$$\mu_1 - \mu_{a_T} = \mu_1(C_0) - \mu_1(C_{L+1}).$$

By applying the telescoping sum property:

$$\mu_1(C_0) - \mu_1(C_{L+1}) = \sum_{l=0}^L (\mu_1(C_l) - \mu_1(C_{l+1})).$$

Using the given condition, we obtain:

$$\sum_{l=0}^L (\mu_1(C_l) - \mu_1(C_{l+1})) \leq \sum_{l=0}^L p_l\epsilon \leq \epsilon.$$

Thus, the event

$$\bigcap_{l=0}^L (\mu_1(C_l) - \mu_1(C_{l+1}) \leq p_l\epsilon)$$

is a subset of the event

$$(\mu_1 - \mu_{a_T} \leq \epsilon).$$

Applying *De Morgan's law*, we obtain the following relation:

$$(\mu_1 - \mu_{a_T} > \epsilon) \subset \bigcup_{l=0}^L (\mu_1(C_l) - \mu_1(C_{l+1}) > p_l\epsilon). \tag{10}$$

**Probability Bound:**

The relation in (10) implies that

$$\mathbb{P}\left(\mu_1 - \mu_{a_T} > \epsilon\right) \le \sum_{l=0}^{L} \mathbb{P}\left(\mu_1(C_l) - \mu_1(C_{l+1}) > \epsilon \cdot p_l\right).$$

Applying the result of bound, we get:

$$\mathbb{P}\left(\mu_1 - \mu_{a_T} > \epsilon\right) \le 3 \sum_{l=0}^{L} \exp\left(-\frac{\epsilon^2}{8} p_l^2 t_0 T_l\right).$$

When $T_l = r^l$, we set:

$$p_l = \left(1 - r^{-0.5}\right)^2 \cdot (l+1) \cdot r^{-0.5l}.$$

Then $\sum_{l=0}^{\infty} p_l = 1$ and $p_l^2 r^l = (1 - r^{-0.5})^4 (l+1)^2 r^{-l+l}$.

Hence,

$$\mathbb{P}(\mu_1(C_l) - \mu_1(C_{l+1}) > p_l\epsilon) \le 3\exp(-K\epsilon^2 t_0(l+1)^2).$$

### 3. The order of $t_0$

If $r \in (1, 2)$ for $T_l = r^l$, then the required budget is

$$N(\lceil t_0 \rceil + \lceil t_0 r \rceil/2 + \ldots + \lceil t_0 r^L \rceil/2^L) \le N((t_0 + 1) + (t_0 r + 1)/2 + \ldots + (t_0 r^L + 1)/2^L)$$

$$\le Nt_0 \left(\frac{1 - (r/2)^L}{1 - r/2}\right) + N\left(2 - 1/2^L\right)$$

$$\le N(t_0 + 2)/(1 - r/2)$$

and $T_l = (l+1)r^l$, the required budget is less than or equal to

$$N(t_0 + 2)/(1 - r/2)^2.$$

To ensure the budget constraint is satisfied, one of the following conditions must hold:

1. If $T_l = r^l$, the constraint is:

$$N\left(\lceil t_0 \rceil + \frac{\lceil t_0 r \rceil}{2} + \frac{\lceil t_0 r^2 \rceil}{2^2} + \cdots + \frac{\lceil t_0 r^L \rceil}{2^L}\right) \le T,$$

2. If $T_l = (l+1)r^l$, the constraint is:

$$N\left(\lceil t_0 \rceil + \frac{\lceil 2 \cdot t_0 r \rceil}{2} + \frac{\lceil 3 \cdot t_0 r^2 \rceil}{2^2} + \cdots + \frac{\lceil (L+1) \cdot t_0 r^L \rceil}{2^L}\right) \le T.$$

In either case, it is sufficient to set:

$$t_0 = const \cdot \frac{T}{N}.$$

$\square$

**Lemma D.11.** *Consider the SH algorithm with a budget schedule $T_l = \lceil (l+1)r^l \rceil$ for $r_0 \in (1, 2)$. The misidentification probability is bounded by:*

$$\mathbb{P}(\mu_1 - \mu_{a_T} > \epsilon) \le \exp\left\{-const \cdot \left(\frac{T\epsilon^2}{N}\right)\right\}.$$

*Proof.* If $T_l = (l+1)r^l$, we can get the similar upper bound

$$3\left(\frac{1}{\sqrt{K_2\epsilon^2 t_0}} + 1\right) \cdot \exp\left\{-K_2\epsilon^2 t_0\right\}$$

where $K_2 = \frac{(1-r^{-0.5})^2}{8}$

by setting

$$p_l = \left(1 - r^{-0.5}\right) r^{-0.5l}.$$

Here, $t_0 = \frac{T}{N}(1-r/2)^2 - 2$ when $T_l = (l+1)r^l$.

$\square$

**Lemma D.12.** *Suppose we apply SH with budget schedule $C_1 \cdot l^{4+\eta} \leq T_l \leq C_2 \cdot r^l$ for some $C_1, C_2 > 0, 1 < r < 2$ and $\eta > 0$. Then*

$$\mathbb{P}(\mu_1 - \mu_{a_T} > \epsilon) \leq \exp\left\{-const \cdot \left(\frac{T\epsilon^2}{N}\right)\right\}$$

.

*Proof.* As we have shown in Theorem 4.10, for $p_l$ such that $\sum_{l=0}^{L} p_l \leq 1$,

$$\mathbb{P}\left(\mu_1 - \mu_{a_T} > \epsilon\right) \leq \sum_{l=0}^{L} \mathbb{P}\left(\mu_1(C_l) - \mu_1(C_{l+1}) > \epsilon \cdot p_l\right).$$

As we set $p_l = \frac{\zeta(1+\eta/3)}{l^{1+\eta/3}}$ where $\zeta(x = \sum_n n^{-x}$, we have upper bound of $\mathbb{P}\left(\mu_1(C_l) - \mu_1(C_{l+1}) > \epsilon \cdot p_l\right)$ as

$$3\exp\left\{-\frac{\epsilon^2}{8}p_l^2 t_0 T_l\right\} = 3\exp\left\{-\frac{\epsilon^2\zeta(1+\eta/3)^2}{8}l^{2+2\eta/3}t_0 T_l\right\} \leq 3\exp\left\{-\frac{\epsilon^2\zeta(1+\eta/3)^2}{8}t_0 l^2\right\}.$$

By considering $l \in \{0, 1, \ldots, L\}$ and substituting $t_0$ with $N$ when $T_l$ is strictly less than $2^l$ (up to a constant factor), or with $N\log_2 N$ when $T_l = \Theta(2^l)$, we obtain a similar upper bound for $T_l = r^l$ with $r \in (1, 2)$.

$\square$

### D.3.2. PROPOSITION 4.11

As described in Section D.3.1, we provide the optimal budget allocation rules for both cases: $T_l = r^l$ and $T_l = (l+1)r^l$. Here, the proof requires the exact formula for the upper bound as described in Theorem 4.10, rather than its big-O approximation.

**Proposition D.13.** *Suppose $T = kN$ for some constant $k$ such that $t_0 T_0 \geq 1$, where*

$$t_0 = \text{argmax}\{t : tN \cdot (T_0 + T_1/2 + \ldots + T_L/2^L) \leq T\}.$$

1. ***For the budget schedule $T_l = r^l$ with $r \in (1, 2)$:***

   *Setting $r_0$ as the solution to the equation $r_0 + r_0^{1.5} - 4 = 0$ (with $r_0 \approx 1.728$) achieves a near-optimal upper bound.*

2. ***For the budget schedule $T_l = (l+1)r^l$ with $r \in (1, 2)$:*** *Setting $r_0$ as the solution to the equation $r_0 - 2r_0^{1.5} + 2 = 0$ (with $r_0 \approx 1.434$) achieves a near-optimal upper bound.*

*proof of $T_l = r^l$.* Note that the bound is a decreasing function of $K = t_0(1 - r^{-0.5})^4$, but $(r, t_0)$ must satisfy the following budget constraint:

$$\frac{1}{2^0} \cdot \lceil t_0 r^0 \rceil + \frac{1}{2^1}\lceil t_0 r^1 \rceil + \ldots + \frac{1}{2^L}\lceil t_0 r^L \rceil \leq const.$$

Since $\lceil x \rceil \le x + 1$, the required budget is less than

$$\frac{1}{2^0} \cdot (t_0 r^0 + 1) + \frac{1}{2^1}(t_0 r^1 + 1) + \ldots + \frac{1}{2^L}(t_0 r^L + 1) = t_0 \frac{1 - (r/2)^L}{1 - r/2} + 2 - 1/2^L. \tag{11}$$

This implies the constraint that

$$t_0/(1 - r/2) \le const.$$

Using Lagrangian Multiplier technique, we can define

$$\mathcal{L}(r, t_0, \lambda) = 4 \ln(1 - r^{-0.5}) + \ln t_0 + \lambda \left(const + \ln t_0 - \ln(1 - r/2)\right).$$

Here, $4 \ln(1 - r^{-0.5}) + \ln t_0$ is the $\log K$.

As differentiating $\mathcal{L}$, we get

$$\frac{\partial \mathcal{L}}{\partial t_0} = 0 \quad \Longleftrightarrow \quad \frac{1}{t_0} + \lambda \frac{1}{t_0} = 0 \qquad \Longleftrightarrow \quad \lambda = -1,$$

$$\frac{\partial \mathcal{L}}{\partial r} = 0 \quad \Longleftrightarrow \quad -r - r^{1.5} + 4 = 0 \quad \Longleftrightarrow \quad r_0 \approx 1.728,$$

$$\frac{\partial \mathcal{L}}{\partial \lambda} = 0 \quad \Longleftrightarrow \quad \frac{t_0}{1 - r/2} = const \quad \Longleftrightarrow \quad t_0 = const \cdot \left(1 - \frac{r_0}{2}\right).$$

Since $r$ does not depend on $t_0$ or $T$, we can set $\hat{r}$ first and then $t_0$ corresponding to the constraint with selected $\hat{r}$. $\qquad \square$

*proof of $T_l = (l+1)r^l$.* Similar to $T_l = r^l$, we should maximize $g_2(r, t_0) = 2 \ln(1 - r^{-0.5}) + \ln t_0$ constrained to the budget bound

$$t_0/(1 - r/2) \le const.$$

Applying the same argument to $g_2$ and budget bound, we can define $\mathcal{L}(r, t_0, \lambda)$ as

$$\mathcal{L}(r, t_0, \lambda) = 2 \ln(1 - r^{-0.5}) + \ln t_0 + \lambda \left(const + \ln t_0 - 2 \ln(1 - r/2)\right).$$

and obtain $\hat{r}$ as a solution of

$$\frac{\partial \mathcal{L}}{\partial r} = 0 \quad \Longleftrightarrow \quad r - 2r^{1.5} + 2 = 0 \quad \Longleftrightarrow \quad r_0 \approx 1.434,$$

$$\frac{\partial \mathcal{L}}{\partial \lambda} = 0 \quad \Longleftrightarrow \quad \frac{t_0}{1 - r/2} = const \quad \Longleftrightarrow \quad t_0 = const \cdot \left(1 - \frac{r_0}{2}\right).$$

$$\square$$

### D.4. Proposition 4.12

**Corollary D.14.** *Under the data-poor condition, the probability of misidentifying the $\epsilon/2$-best arm within the candidate set $C$,*

$$\mathbb{P}(\mu_1(C) - \mu_{a_T} > \epsilon/2)$$

*is bounded by:*

- *US:* $\le N \cdot \exp\left\{-C_1 \cdot \left(\frac{T\epsilon^2}{N}\right)\right\}$
- *BSH:* $\le \exp\left\{-C_2 \cdot \left(\frac{T\epsilon^2}{N(\ln T)^2}\right)\right\}$
- *B3:* $\le \exp\left\{-C_3 \cdot \left(\frac{T\epsilon^2}{N}\right)\right\}$,

*for some constants $C_1, C_2, C_3 > 0$.*

*Proof.* **Proof of US:** $\leq N \cdot \exp\left\{-C_1\left(\frac{T\epsilon^2}{N}\right)\right\}$

We define $t_0$ as the number of pulls per arm in US, and let $\hat{\mu}_i$ denote the average reward of arm $i$ after $t_0$ pulls.

Suppose the average reward of best arm satisfies

$$(\hat{\mu}_1 > \mu_1 - \epsilon/2)$$

and all arms which are not $\epsilon/2$-best arm also satisfy

$$(\hat{\mu}_i \leq \mu_i + \epsilon/2).$$

Then $\hat{\mu}_1 - \hat{\mu}_i > (\mu_1 - \epsilon/2) - (\mu_i + \epsilon/2) = (\mu_1 - \mu_i) - \epsilon > 0$, thus the non-$\epsilon/2$ best arms cannot be selected as the estimated best arm. Specifically,

$$(\mu_1 - \mu_{a_T} > \epsilon) \subset \left((\hat{\mu}_1 > \mu_1 - \epsilon/2) \cap \bigcap_{i:\mu_1-\mu_i>\epsilon}(\hat{\mu}_i \leq \mu_i + \epsilon/2)\right)^c,$$

or more generally,

$$(\hat{\mu}_1 < \mu_1 - \epsilon/2) \cup \left(\bigcup_{i\neq 1}(\hat{\mu}_i > \mu_i + \epsilon/2)\right).$$

$(\because \{i : \mu_1 - \mu_i > \epsilon\} \subset \{i = 2, 3, \ldots, N\})$

Using the Hoeffding's bound, the probability of each event is bounded as:

$$\mathbb{P}(\hat{\mu}_i > \mu_i + \epsilon/2) \quad \text{or} \quad \mathbb{P}(\hat{\mu}_i < \mu_i - \epsilon/2) \leq \exp\left(-\frac{t_0\epsilon^2}{8}\right).$$

Since $t_0 = \Theta(T/N)$, we have:

$$\mathbb{P}(\mu_1 - \mu_{a_T} > \epsilon) \leq \mathbb{P}(\hat{\mu}_1 < \mu_1 - \epsilon/2) + \sum_{i\neq 1}\mathbb{P}(\hat{\mu}_i > \mu_i + \epsilon/2) \leq N\exp\left(-\frac{T\epsilon^2}{8N}\right).$$

**Proof of BSH:** $\leq \exp\left\{-C_2\left(\frac{T\epsilon^2}{N(\ln T)^2}\right)\right\}$

Suppose there are $B$ brackets within the budget $T$ and $B_C$ is the number of brackets in the candidate set $C$. Then the event that $\epsilon$-best arm within set $C$ is misidentified is a subset of the event that there exists a bracket which fails to find $\epsilon$-best arm within set $C$:

$$(\mu_1(C) - \mu_{a_T} > \epsilon)$$
$$\subset \bigcup_{b \leq B_C}(\mu_1(\{i \in \text{bracket } b\}) - \mu_{\text{the estimated arm in bracket } b} > \epsilon). \tag{12}$$

The $b^{th}$ bracket has the following upper bound of misidentification probability:

$$\mathbb{P}\left(\mu_1(\{i \in \text{bracket } b\}) - \mu_{\text{the estimated arm in bracket } b} > \epsilon\right)$$
$$\leq \exp\left\{-\Omega\left(\text{the initial budget of bracket } b \text{ at the last round R} \cdot \epsilon^2\right)\right\}.$$

1. The number of brackets, $B$(opened) and $B_C$(brackets in the candidate set)

$$B = \arg\max\{b \mid (b-1)\ln 2 \cdot e^{(b-1)\ln 2} \leq T\}.$$

The **Lambert function**, denoted as $W(x)$, is defined as the inverse function of $x = W(x)e^{W(x)}$, meaning it satisfies:

$$W(x)e^{W(x)} = x.$$

Since $W(x)$ has an asymptotic order of $\ln x$(Hoorfar & Hassani, 2008), we obtain:

$$B = W(T) + 1 \sim \ln T.$$

Similarly, $B_C$ is derived in the proof of Proposition 4.12 as the solution to:

$$\mathcal{B} = \arg\max \left\{ 2^b \mid (b-1)2^{(b-1)} + b^2 2^b \leq T \right\} = \Theta(T/(\log_2 T)^2).$$

From this, we conclude:

$$B_C = \log_2 \mathcal{B} = \Theta(\ln T - 2 \ln \ln T).$$

2. The last round finished $R$ Then the $b^{th}$ bracket is allocated the budget

$$\frac{(b+1)2^{b+1} - b2^b}{b} + \frac{(b+2)2^{b+2} - (b+1)2^{b+1}}{b+1} + \ldots + \frac{T - (B-1)2^{B-1}}{B}.$$

(a) $b = B$

**Claim:** This bracket cannot return the estimated best arm.

$$B_C = \arg\max\{b : (2b^2 + b - 1)2^b \leq T - 1\}.$$

Suppose $B$ satisfy the above condition. Then,

$$(2B^2 + B - 1)2^B \leq T - 1 \leq B2^B - 1.$$

When $B = 1$, $(2B^2 + B - 1)2^B = 2 \times 2^1$ and $B2^B = 2$. Since the LHS grows faster than RHS, this is a contradiction.

(b) $b = B - 1$

As the same argument with $b = B$, the following inequality should be satisfied for bracket $b$ to be in the candidate set $C$:

$$(2(B-1)^2 + (B-1) - 1)2^{B-1} \leq T - 1 \leq B2^B - 1.$$

The above inequality is satisfied only when $B = 2(\because B \geq 2$ to be $b = B - 1 \geq 1)$. When $B = 2$, the allocated budget to $b = B - 1$ equals to

$$\frac{2^B}{B-1} + \frac{T}{B} - 2^{B-1} = \frac{2^{B-1} + T}{B} = \Theta\left(\frac{T}{B}\right)(\because 2^B = \Theta(T)).$$

(c) $b \leq B - 2$

The allocated budget can be further simplified as

$$\sum_{k=b}^{B-2} \frac{(k+1)2^{k+1}}{k \cdot (k+1)} + \frac{T}{B} - \frac{b2^b}{b}$$

$$= \sum_{k=b}^{B-2} \frac{2^{k+1}}{k} + \frac{T}{B} - 2^b$$

$$\geq \frac{2^{b+1}}{B-2} \cdot (2^{B-2-b+1} - 1) + \frac{T}{B} - 2^b$$

$$\underset{(*)}{\gtrsim} \frac{T}{B}$$

The inequality in terms of order in $(*)$ is due to

- $B = \Theta(\ln T)$,
- $B2^b \le B2^B \le T$.

Since each round $r$ starts from $2^r$ and requires $2^r \cdot b2^b$ budget, the round $R$ satisfying the following inequality is guaranteed to be finished:

$$\sum_{r=0}^{R-1} 2^r \cdot b2^b \lesssim T/B \iff (2^R - 1) \lesssim \frac{T}{b2^b \cdot B}.$$

Combining these, we get

$$\mathbb{P}\left(\mu_1(\{i \in \text{bracket } b\}) - \mu_{\text{the estimated arm in bracket } b} > \epsilon/B\right)$$

$$\le \exp\left\{-\Omega\left(\frac{T}{b2^b \cdot B} \cdot \epsilon^2\right)\right\}$$

$$\le \exp\left\{-\Omega\left(\frac{T\epsilon^2}{2^b(\ln T)^2}\right)\right\} . (\because b \le B, B = \Theta(\ln T))$$

Thus,

$$\mathbb{P}(\mu_1(C) - \mu_{a_T} > \epsilon) \le \sum_{b=1}^{B} \exp\left\{-\Omega\left(\frac{T\epsilon^2}{2^b(\ln T)^2}\right)\right\}$$

$$\le \exp\left\{-\Omega\left(\frac{T\epsilon^2}{(\ln T)^2} \cdot \frac{2^b}{N}\right)\right\}$$

The last inequality comes from the fact that $2^b \le N$ under the data-poor condition. Using the technique used in the proof of Theorem 4.10, we get

$$\mathbb{P}(\mu_1(C) - \mu_{a_T} > \epsilon) \le \exp\left\{-\Omega\left(\frac{T\epsilon^2}{N(\ln T)^2}\right)\right\}.$$

**Proof of B3:** $\mathbb{P}(\mu_1 - \mu_{a_T} > \epsilon) \le \exp\left\{-C_3\left(\frac{T\epsilon^2}{N}\right)\right\}$

To provide a rigorous proof, we decompose the analysis into three steps, as established in our budget and screening capacity analysis. We first define:

- $l^*$ : the arm index of the best arm in set $C_l$,
- $C_l(\epsilon)$ : the set of arms in $C_l$ whose true mean is larger than $\mu_1(C_l) - \epsilon$,
- $\text{Box}(l, j; i)$ : the set of arms in $\text{Box}(l, j)$ when arm $i$ is present and the box is full.

As established in Section 4.1, the event $(\mu_1 - \mu_{a_T} > \epsilon)$ is contained within the union of level-wise error events. Specifically, for any sequence $(p_l)_l$ such that $\sum_l p_l \le 1$:

$$(\mu_1 - \mu_{a_T} > \epsilon) \subseteq \bigcup_{l=0}^{L} (\mu_1(C_l) - \mu_1(C_{l+1}) > p_l\epsilon)$$

where

$$C_l \equiv \{i : \text{arm } i \text{ has been in level } l\}$$

**Step 1: Bounding Level-wise Error and Candidate Budget** $T_C$  For the event $(\mu_1(C_l) - \mu_1(C_{l+1}) > p_l\epsilon)$ to occur, arm $l^*$ must fail to be promoted. This happens if:

1. The average reward of $l^*$ is either the median or the minimum in $\text{Box}(l, 0; l^*)$.

2. There exists an arm $i \in \text{Box}(l, 0; l^*)$ such that $i \notin C_l(p_l\epsilon)$.

*Claim* 1. The event $(\mu_1(C_l) - \mu_1(C_{l+1}) > p_l\epsilon)$ implies that the empirical mean of the best arm $l^*$ is not the strictly largest in its box: $\hat{\mu}_{l^*}^l \le \hat{\mu}_{\text{med}[l,0;l^*]}^l$.

*Proof of Claim.* In B3, an arm $i \in C_l$ is only removed or deferred if it fails to be the largest in its box comparison. The suspension mechanism (SHIFT) ensures that even if an arm is the median, it is revisited. However, for the best arm to be lost from the candidate set at level $l$, it must be either discarded (minimum) or suspended indefinitely. In both cases, the initial condition is $\hat{\mu}_{l*}^l \leq \hat{\mu}_{\mathrm{med}[l,0;l*]}^l$.

By applying the concentration inequalities derived in Section D.2, the probability of this error at level $l$ is:

$$\mathbb{P}(\mu_1(C_l) - \mu_1(C_{l+1}) > p_l \epsilon) \leq 3 \exp\left(-\frac{r_0^l p_l^2 \epsilon^2}{4}\right).$$

By setting the sequence $p_l = (1 - r_0^{-0.5})^2 \cdot (l+1) \cdot r_0^{-0.5l}$ and summing over all levels $l$, we obtain a bound as a function of the total pulls allocated to candidate arms, $T_C$:

$$\mathbb{P}(\mu_1 - \mu_{a_T} > \epsilon) \leq \exp\left(-\Omega\left(\frac{T_C \epsilon^2}{N}\right)\right).$$

**Step 2: Proving the Budget Proportionality $T_C = \Theta(T)$**   A critical part of our analysis is ensuring that the candidate budget $T_C$ is not negligible compared to the total budget $T$. As shown in the proof of $c_0 = \Theta(T)$ (Appendix A), the number of initial candidate arms $n_0$ is proportional to $T$. Since B3 allocates a constant base pull $r_0$ and the hierarchical structure ensures that the total number of pulls follows a geometric progression across levels, the total budget spent on processing these $n_0$ arms ($T_C$) maintains the same order as $T$. Formally, there exists a constant $\alpha > 0$ such that:

$$T_C \geq \alpha T.$$

**Step 3: Final Substitution and Result**   Substituting the proportionality $T_C = \Theta(T)$ into the bound obtained in Step 1:

$$\mathbb{P}(\mu_1 - \mu_{a_T} > \epsilon) \leq \exp\left(-\Omega\left(\frac{\alpha T \epsilon^2}{N}\right)\right) = \exp\left(-\Omega\left(\frac{T\epsilon^2}{N}\right)\right).$$

This concludes that the misidentification probability of B3 in the data-poor regime matches the optimal rates up to constant factors, while maintaining its fully anytime property.

$\square$

## D.5. A Proof of Instance-dependent Upper Bound in Theorem 4.3

*Proof.* We decompose the overall failure event into the non-inclusion of the best arm into the initial candidate set $C$, and the subsequent misidentification within $C$:

$$\mathbb{P}(\mu_1 - \mu_{a_T} > \epsilon) \leq \underbrace{\mathbb{P}(\mu_1 \notin C)}_{\text{Non-Inclusion } (\mathbb{P}_{\text{NI}})} + \underbrace{\mathbb{P}(\mu_1 \in C \text{ and } \mu_1 - \mu_{a_T} > \epsilon)}_{\text{Misidentification } (\mathbb{P}_{\text{MI}})}.$$

**Part 1: Bounding the Non-Inclusion Probability ($\mathbb{P}_{\text{NI}}$)**   Under the data-poor condition, the algorithm subsamples a restricted initial candidate set $C \subset [N]$ of size $c_0$. Let $N_{\epsilon/2}$ denote the number of $\epsilon/2$-best arms in the entire universe. By standard combinatorial tail bounds for sampling without replacement, the probability that the initial candidate set $C$ completely misses all $\epsilon/2$-best arms (and hence fails to include the true best arm $\mu_1$) is bounded as:

$$\mathbb{P}_{\text{NI}} = \mathbb{P}(\mu_1 \notin C) \leq \exp\left(-c\frac{c_0 N_{\epsilon/2}}{N}\right)$$

for some absolute constant $c > 0$. To express this upper bound in terms of the instance complexity $H_2(\epsilon)$, we utilize the definition:

$$H_2(\epsilon) := \frac{1}{N_{\epsilon/2}} \max_{i:\Delta_i \geq \epsilon} \frac{i}{\Delta_i^2}.$$

By our standard assumption on the boundary instance scaling $\Delta_N \asymp 1$, substituting the largest arm index $i = N$ into the maximum yields the lower bound:

$$H_2(\epsilon) \geq \frac{1}{N_{\epsilon/2}} \cdot \frac{N}{\Delta_N^2} \geq \frac{N}{N_{\epsilon/2}},$$

which directly implies the fundamental ratio relation:

$$\frac{N_{\epsilon/2}}{N} \geq \frac{1}{H_2(\epsilon)}.$$

Substituting this relation into the exponent of the sub-sampling tail bound, we obtain:

$$\exp\left(-c\frac{c_0 N_{\epsilon/2}}{N}\right) \leq \exp\left(-c\frac{c_0}{H_2(\epsilon)}\right).$$

Finally, we link the initial candidate size $c_0$ to the total budget $T$ and the maximum hierarchy level $L$ under the budget allocation rules of B3. When $r_0 < 2$, the configuration budget allocation ensures a robust early-stage exploration where $c_0 \asymp T \geq c'\frac{T}{L}$ for some constant $c' > 0$. Consequently, for both budget growth rates, the initial level size is bounded below by $\Omega(T/L)$. This establishes the desired exponential decay for the non-inclusion probability uniformly across all regimes:

$$\mathbb{P}_{\text{NI}} \leq \exp\left(-C_{NI}\frac{T}{L \cdot H_2(\epsilon)}\right),$$

where $C_{NI} > 0$ is a universal positive constant independent of $T, N$, and $\epsilon$.

**Part 2: Bounding the Misidentification Probability ($\mathbb{P}_{\text{MI}}$)** **Step 1: Decomposition by the first elimination level of all $\varepsilon/2$-best arms.**

For each level $\ell$, define

$$\mathcal{B}_\ell(\varepsilon/2) = \{i \in C_\ell : \Delta_i \leq \varepsilon/2\},$$

and let

$$G_\ell = \{\mathcal{B}_\ell(\varepsilon/2) = \mathcal{B}_{\ell+1}(\varepsilon/2)\}.$$

Thus $G_\ell$ denotes the event that level $\ell + 1$ still contains the all $\varepsilon/2$-best arm.

Define the first elimination level by

$$L^* = \min\{\ell \geq 1 : G_{\ell-1} \text{ holds but } G_\ell \text{ fails}\}.$$

Equivalently, level $L^*$ is the first level at which at least one of the $\varepsilon/2$-best arm is eliminated.

Then

$$\{L^* = \ell\} = G_\ell^c \cap \bigcap_{k<\ell} G_k.$$

If

$$\mu_1 - \mu_{\hat{a}_T} > \varepsilon,$$

then the final selected arm is non-$\varepsilon$-best. Hence all $\varepsilon/2$-best arms must have lost at some level. Therefore

$$\{\mu_1 - \mu_{\hat{a}_T} > \varepsilon\} \subseteq \bigcup_\ell \left(G_\ell^c \cap \bigcap_{k<\ell} G_k\right).$$

Consequently,

$$\{\mu_1 - \mu_{\hat{a}_T} > \varepsilon\} \subseteq \bigcup_\ell \left(G_\ell^c \cap \bigcap_{k<\ell} G_k\right) \cap \{\mu_1 - \mu_{a_T}\}.$$

Equivalently,

$$\mathbb{P}(\mu_1 - \mu_{\hat{a}_T} > \varepsilon) \leq \sum_\ell \mathbb{P}\left(G_\ell^c \quad \text{and} \quad \mu_1 - \mu_{\hat{a}_T} > \varepsilon \,\middle|\, \bigcap_{k<\ell} G_k\right).$$

**Step 2: Amplification through independent beating branches.** Fix a decomposition level $\ell$ and work on the event

$$G_\ell^c \cap \{\mu_1 - \mu_{\hat{a}_T} > \varepsilon\} \cap \bigcap_{k<\ell} G_k.$$

By the definition of $\ell$ as the first level at which an $\varepsilon/2$-best arm is eliminated, no $\varepsilon/2$-best arm has been eliminated before level $\ell$. Therefore, at level $\ell$, the number of remaining $\varepsilon/2$-best arms is still

$$N_{\varepsilon/2} := \left|\{i : \Delta_i \leq \varepsilon/2\}\right|.$$

On the event $\{\mu_1 - \mu_{\hat{a}_T} > \varepsilon\}$, the final recommended arm is non-$\varepsilon$-best. Hence every $\varepsilon/2$-best arm present at level $\ell$ must eventually be beaten along a sequence of comparisons that terminates at a non-$\varepsilon$-best arm. More precisely, for each such arm $a$, we can construct a beating path

$$a = b_0 \to b_1 \to \cdots \to b_K,$$

where $b_{s+1}$ is the arm that beats $b_s$ at the corresponding later comparison. The path satisfies

$$\Delta_{b_0} \leq \varepsilon/2, \qquad \Delta_{b_K} > \varepsilon.$$

If $b_s$ is promoted to the next level without being eliminated at that stage, we set $b_{s+1} = b_s$ and assign the corresponding gap $d_s = 0$.

Define

$$d_s = (\mu_{b_s} - \mu_{b_{s+1}})_+, \qquad s = 0, \ldots, K-1.$$

Since the path starts from an $\varepsilon/2$-best arm and terminates at a non-$\varepsilon$-best arm,

$$\sum_{s=0}^{K-1} d_s \geq \mu_{b_0} - \mu_{b_K} = \Delta_{b_K} - \Delta_{b_0} > \Delta_{N_{\varepsilon.2}}.$$

We now use the branching structure of Box Thirding. Let $c_j$ denote the number of arms at level $j$, and define the matching ratio

$$\rho_j := \frac{c_j - c_{j+1}}{c_{j+1}}.$$

This is the number of eliminated arms per promoted arm when passing from level $j$ to level $j+1$. Since Box Thirding promotes a fraction between $1/3$ and $1/2$ of the current level, we have

$$1 \leq \rho_j \leq 2.$$

Moreover, except possibly at the last level, the promoted fraction is strictly larger than $1/3$. Hence

$$\rho_j < 2$$

for all non-terminal levels $j$. Therefore, for the levels relevant to the beating path, there exists a constant $\rho < 2$ such that

$$\rho_j \leq \rho < 2, \qquad \text{for all } j < L.$$

At relative depth $s = 0$, the $N_{\varepsilon/2}$ surviving $\varepsilon/2$-best arms can be matched to at least

$$\frac{N_{\varepsilon/2}}{\rho_\ell}$$

independent beating arms. Recursively, at relative depth $s$, the number of independent charged beating events is at least

$$\frac{N_{\varepsilon/2}}{\rho_\ell \rho_{\ell+1} \cdots \rho_{\ell+s}}.$$

Using the uniform bound $\rho_j \leq \rho$, this is bounded below by

$$const \cdot \frac{N_{\varepsilon/2}}{\rho^{s+1}}.$$

A reversal at relative depth $s$ occurs at level $\ell + s$, where the sample size is of order $r_0^{\ell+s}$. Hence the sub-Gaussian reversal bound gives

$$\mathbb{P}(\text{one reversal at relative depth } s) \leq \exp\{-cr_0^{\ell+s}d_s^2\}.$$

Since at least $N_{\varepsilon/2}/\rho^{s+1}$ such events can be selected independently at relative depth $s$, the probability of the required beating structure is bounded by

$$\prod_{s=0}^{K-1} \left[\exp\{-cr_0^{\ell+s}d_s^2\}\right]^{N_{\varepsilon/2}/\rho^{s+1}}.$$

Equivalently,

$$\exp\left\{-c\sum_{s=0}^{K-1} r_0^{\ell+s}d_s^2 \frac{N_{\varepsilon/2}}{\rho^{s+1}}\right\} = \exp\left\{-\frac{c}{\rho}r_0^{\ell}N_{\varepsilon/2}\sum_{s=0}^{K-1}\left(\frac{r_0}{\rho}\right)^s d_s^2\right\}.$$

Finally, by weighted Cauchy–Schwarz,

$$\left(\sum_{s=0}^{K-1} d_s\right)^2 \leq \left(\sum_{s=0}^{K-1}\left(\frac{\rho}{r_0}\right)^s\right)\left(\sum_{s=0}^{K-1}\left(\frac{r_0}{\rho}\right)^s d_s^2\right).$$

Since $\sum_s d_s > \varepsilon/2$,

$$\sum_{s=0}^{K-1}\left(\frac{r_0}{\rho}\right)^s d_s^2 \geq \frac{\Delta_{N_{\varepsilon/2}}^2/4}{\sum_{s=0}^{K-1}(\rho/r_0)^s}.$$

Since

$$\sum_{s=0}^{K-1} d_s > \varepsilon/2,$$

weighted Cauchy–Schwarz yields

$$\left(\sum_{s=0}^{K-1} d_s\right)^2 \leq \left(\sum_{s=0}^{K-1}\left(\frac{\rho}{r_0}\right)^s\right)\left(\sum_{s=0}^{K-1}\left(\frac{r_0}{\rho}\right)^s d_s^2\right).$$

Because $r_0 > \rho$, the geometric series

$$\sum_{s=0}^{K-1}\left(\frac{\rho}{r_0}\right)^s$$

is uniformly bounded by a constant depending only on $r_0/\rho$. Therefore there exists a constant $C > 0$ such that

$$\sum_{s=0}^{K-1}\left(\frac{r_0}{\rho}\right)^s d_s^2 \geq C\Delta_{N_{\varepsilon/2}}^2.$$

Consequently,

$$\mathbb{P}\left(G_\ell^c \cap \{\mu_1 - \mu_{\hat{a}_T} > \varepsilon\}\,\Bigg|\,\bigcap_{k<\ell} G_k\right) \leq \exp\{-const \cdot N_{\varepsilon/2} \cdot r_0^{\ell}\Delta_{\varepsilon/2}^2\}.$$

**Step 3: Decomposition within level $\ell$.**

Now set

$$c^* := \frac{N_{\varepsilon/2}}{N}.$$

Since $c_\ell \leq N$, we have

$$N_{\varepsilon/2} = c^*N \geq c^*c_\ell.$$

Therefore the bound from Step 2 implies

$$\mathbb{P}\left(G_\ell^c \cap \{\mu_1 - \mu_{\hat{a}_T} > \varepsilon\}\,\Bigg|\,\bigcap_{k<\ell} G_k\right) \leq \exp\{-const \cdot N_{\varepsilon/2} \cdot r_0^{\ell}\Delta_{\varepsilon/2}^2\}.$$

Equivalently, using the definition

$$H_2(\varepsilon) = \frac{1}{N_{\varepsilon/2}} \cdot \sup_{i:\Delta_i \geq \varepsilon} \frac{i}{\Delta_i^2},$$

and the fact that $N_{\varepsilon/2} \geq c^* c_\ell$, this may be written in the instance-dependent form

$$\mathbb{P}\left( G_\ell^c \cap \{\mu_1 - \mu_{\hat{a}_T} > \varepsilon\} \,\middle|\, \bigcap_{k<\ell} G_k \right) \leq \exp\left\{ -c\frac{r_0^\ell c^* c_\ell}{H_2(\varepsilon)} \right\} = \exp\left\{ -const \cdot \frac{r_0^\ell c_\ell}{H_2(\varepsilon)} \right\}.$$

**Step 4: Summation over $\ell$.**

It remains to lower bound the level-wise exponent uniformly over $\ell$.

When $r_0 < 2$, the product $r_0^\ell c_\ell$ decreases with $\ell$, since

$$c_\ell \asymp \frac{c_0}{2^\ell}$$

and hence

$$r_0^\ell c_\ell \asymp c_0 \left(\frac{r_0}{2}\right)^\ell.$$

The early levels are therefore controlled directly by the budget allocation. Since the total budget is split across $L$ levels, the initial-level exponent is at least of order

$$\frac{T}{L}.$$

For later levels, the bound based on $c^* c_\ell$ becomes too conservative, because $c_\ell$ decreases with $\ell$. Instead, we use the sharper quantity $N_{\varepsilon/2}$. Since $N_{\varepsilon/2}$ is fixed across levels, the exponent

$$r_0^\ell N_{\varepsilon/2}$$

increases exponentially with $\ell$. Thus, although $r_0^\ell c_\ell$ decreases, the alternative exponent $r_0^\ell N_{\varepsilon/2}$ grows with $\ell$.

Combining the two regimes, the level-wise exponent is bounded below by

$$r_0^\ell \max\{c^* c_\ell, \, N_{\varepsilon/2}\} \gtrsim \frac{T}{L}.$$

Equivalently, the previous-step probability bound yields

$$\mathbb{P}\left( G_\ell^c \cap \{\mu_1 - \mu_{\hat{a}_T} > \varepsilon\} \,\middle|\, \bigcap_{k<\ell} G_k \right) \leq \exp\left\{ -c\frac{T}{L\,H_2(\varepsilon)} \right\}.$$

Finally, summing over at most $L$ possible first-elimination levels gives

$$\mathbb{P}(\mu_1 - \mu_{\hat{a}_T} > \varepsilon) \leq L \exp\left\{ -c\frac{T}{L\,H_2(\varepsilon)} \right\}.$$

**Conclusion: Final Assembly** Combining the bounds for $\mathbb{P}_{\mathrm{NI}}$ and $\mathbb{P}_{\mathrm{MI}}$, and noting that under the data-poor horizon $(T \leq N)$, the maximum hierarchy depth scales strictly as $L \leq \log_2 N$, the pre-factor $L$ is seamlessly bounded by $\log_2 N$. This yields the final tight instance-dependent upper bound:

$$\mathbb{P}(\mu_1 - \mu_{a_T} > \epsilon) \leq \log_2 N \cdot \exp\left( -C\frac{T}{\log_2 N \cdot H_2(\epsilon)} \right),$$

where $C > 0$ is a clean, universal positive constant independent of $T$, $N$, and the instance gaps. $\qquad\square$

