# OpenReview forum: "Box Thirding: Anytime Best Arm Identification under Insufficient Sampling"
_ICML.cc/2026/Conference — ICML 2026 regular_

### Official Review · Reviewer_JaAf · 2026-02-25

**Soundness:** 3
**Presentation:** 3
**Significance:** 3
**Originality:** 4
**Overall Recommendation:** 5
**Confidence:** 5

**Summary:**

The paper investigates the anytime best arm identification problem, which differs from the traditional fixed-confidence and fixed-budget BAI settings. The authors propose a novel algorithm that departs from standard elimination-based or confidence-bound-based approaches. Additionally, the paper provides a theoretical analysis of the worst-case identification error bound. Empirical results demonstrate significant improvements over existing state-of-the-art methods.

**Compliance With Llm Reviewing Policy:**

Affirmed.

**Final Justification:**

The rebuttal addressed my main concerns.

**Key Questions For Authors:**

1. Can the authors provide more intuition or discussion on the motivation behind this specific algorithm design? In particular, it would be helpful to understand why a design that moves away from traditional elimination or UCB-based styles was chosen.
2. What is the key technical difficulty that prevents the derivation of an instance-dependent bound similar to those found in the BUCB and BSH papers?
3. Regarding the scope of the results, does the proposed framework and analysis also cover the standard best arm identification problem?
4. In the BSH paper, the authors analyze performance on specific instances such as the "polynomial instance." How do the theoretical results of this paper behave when applied to those same types of instances?

**Limitations:**

The paper has a noticeable limitation regarding the depth of its theoretical analysis. Specifically, the authors do not discuss the lack of an instance-dependent bound, which is a standard expectation in this field, especially since the benchmarks used for comparison (BUCB and BSH) provide such tight, instance-specific guarantees.

**Strengths And Weaknesses:**

Strengths

1. The algorithm design is highly novel. It is refreshing to see a new approach that moves beyond the traditional elimination or UCB-style frameworks that dominate the existing literature.
2. The empirical results are very strong. The authors compare their method against appropriate and competitive benchmarks, including BUCB and BSH.

Weaknesses

1. The theoretical analysis only provides a worst-case bound. This is a significant limitation compared to the BUCB and BSH papers, which both offer strong, tight, instance-dependent bounds for various settings.
2. The approach does not appear to generalize well to the standard best arm identification case.

---

> ### Author Rebuttal · Authors · 2026-03-30
>
> ### Q1: intuition behind the algorithm
>
> We thank the reviewer for this question. Sequential hypothesis testing allows a decision — reject, accept, or defer — to be made at any point in time based solely on the information accumulated so far, without requiring a pre-specified sample size. We view anytime BAI as conceptually analogous: the algorithm must maintain a valid recommendation at every stopping point, using only the pulls observed up to that moment. This connection naturally suggests that incorporating a deferment option — withholding judgment on under-observed arms rather than forcing a binary eliminate/survive decision — should improve performance in the data-poor regime, where premature commitment is the primary source of error. B3 is designed with this philosophy in mind.
>
>
> ### Q2: instance-dependent bound
>
> We note that the B3 instance-dependent bound follows cleanly by combining our existing proof with that of Zhao et al. (2023), as shown below. We initially omitted it not because the derivation is difficult, but because a meaningful comparison with BSH would additionally require reconstructing the logarithmic factors absorbed in BSH's $\widetilde{\Theta}(\cdot)$ guarantees. We agree with the reviewer that the B3 bound is valuable in its own right and will include it in the revision.
>
>
> **Theorem (Instance-Dependent Bound).**
> $$P(\mu_1 - \mu_{a_T} > \varepsilon) \leq \exp\left(-\Omega\left(\frac{T}{H_2(\varepsilon/2)}\right)\right),$$
> *where $H_2(\varepsilon) := \frac{1}{N_{\varepsilon/2}} \cdot \max_{i \geq N_{\varepsilon/2}+1} \frac{i}{\Delta_i^2}$.*
>
> **Proof sketch.**
>
> *(1) Non-inclusion probability.*
>
> $$P(\mu_1 - \mu^*(C) > \varepsilon/2) \leq \exp\left(-\Omega\left(\frac{T N_{\varepsilon/2}}{N}\right)\right).$$
>
> By definition of $H_2(\varepsilon)$, taking $i = N$ gives $H_2(\varepsilon) \geq \frac{N}{N_{\varepsilon/2} \Delta_N^2}$.
>
>
> *(2) Misidentification within $C$.* Let $0 \leq p_1 \leq \cdots \leq p_L = 1/2$ and define
>
> $$G_\ell = (\text{at least } \tfrac{g}{2L}(1-p_\ell)\text{ arms with } \mu_i > \mu_1 - (1-p_\ell)\varepsilon/2 \text{ survive in } C \text{ through level } \ell).$$
>
> Then $\cap_\ell G_\ell \subset (\mu^\ast(C) - \mu_{a_T} \leq \varepsilon/2)$,
> so $P(\mu^*(C) - \mu_{a_T} > \varepsilon/2) \leq \sum_\ell P(G_{\ell+1}^c \mid G_\ell))$.
>
> To $G_{\ell+1}$ be happend, non-$\varepsilon/2$-best arms should remain at least $n_{\ell + 1}$, implying that $\Delta \geq \Delta_{n_\ell} \geq \varepsilon/2$. Applying the definition of $H_2$ at index $i = n_\ell$ gives
> $$\Delta^2 \geq \Delta_{n_\ell}^2 \geq \frac{n_\ell}{N_{\varepsilon/2}H_2(\varepsilon)}.$$
> For $G_{\ell+1}^c \mid G_\ell$ to occur, there must exist $i,j$ such that $\mu_j \leq \mu_1 - \varepsilon/2$ and $\hat\mu_i < \hat\mu_j$ (so $\mu_i - \mu_j \geq p_\ell \Delta/2$ by the gap bound above), and this must happen independently at least $g/(2L)$ times:
> $$P(G_{\ell+1}^c \mid G_\ell) \leq \exp\left(-\Omega\left(\frac{r_0^\ell p_\ell^2 n_\ell}{N_{\varepsilon/2} L H_2(\varepsilon)}\right)\right).$$
> Using $n_\ell \geq c_0/3^\ell$  and setting $p_\ell^2 \propto (l+1)^2 L (3/r_0)^\ell$, and summing over levels by the same argument as Appendix D.4 yields $\exp(-\Omega(T/H_2(\varepsilon)))$. The global $H_2$ follows from $H_2^C \leq 64H_2^{\text{all}}$ (Lemma 13.1, Zhao et al., 2023).
>
>
> ### Q3: Generalization of proposed framework
> Yes, our framework also covers the standard BAI setting. When T is sufficiently large (data-rich regime), the non-inclusion probability becomes zero by construction — every ε-best arm is guaranteed to appear in the candidate set C. In this case, the total error reduces entirely to the misidentification probability within C, for which our bound provides a general upper bound. Thus, the proposed framework subsumes the standard BAI problem as a special case, and the theoretical guarantees remain valid across both regimes.
>
>
> ## Question 4: Performance on Polynomial Gap Instances
>
> By Corollary 1.1 of Zhao et al. (2023), for Polynomial(α) instances with α > 1/2, we have H₂(ε) < 4/ε². Substituting this into our instance-dependent upper bound for B3 of the form exp(−Ω(T/H₂(ε))), we obtain
>
> $$P(\mu_{J_T} < \mu_1 - \varepsilon) \leq \exp\left(-\Omega(T\varepsilon^2)\right),$$
>
> **with no hidden logarithmic factors**. By Corollary 6.2 of Zhao et al. (2023), BSH is shown to match the lower bound on Polynomial(α) instances, establishing exp(−Õ(Tε²)) as the optimal rate for this instance class — however, only up to logarithmic factors. Therefore, B3 not only matches this optimal rate on polynomial gap instances, but does so **without the logarithmic factors present in BSH's bound**.

---

> > ### Author Rebuttal · Reviewer_JaAf · 2026-04-02
> >
> > The authors addressed my main questions. And I'd like to reiterate my initial comment that the algorithm is novel and refreshing for BAI topics, as a conference paper's scope. Thereby I have increased my rating.

---

> > > ### Author Response · Authors · 2026-04-02
> > >
> > > We sincerely thank you for the positive feedback and for increasing the rating.
> > >
> > > We are very encouraged that you found our algorithm novel and refreshing for BAI topics.
> > > We especially appreciate your insightful comments regarding the missing parts of the original manuscript.
> > > Your feedback has been invaluable in helping us identify areas for further refinement.
> > > We will ensure that these improvements and a more rigorous presentation of our theoretical results are fully incorporated into the camera-ready version to deliver a more complete and polished paper.
> > >
> > > Thank you again for your time and constructive guidance.

---

### Official Review · Reviewer_6WaX · 2026-03-09

**Soundness:** 3
**Presentation:** 2
**Significance:** 2
**Originality:** 3
**Overall Recommendation:** 4
**Confidence:** 4

**Summary:**

This paper considers best arm identification under the fixed-budget and anytime setup, and mainly focuses on the data-poor regime. It devises the box-thirding algorithm (B3), and provides theoretical guarantees on the simple regret. Empirical studies are conducted to illustrate the effectiveness of the proposed method.

**Compliance With Llm Reviewing Policy:**

Affirmed.

**Final Justification:**

The proposed algorithm is novel and shows good empirical performance. The authors made great efforts during the rebuttal to show an instance-dependent bound.

It is highly appreciated if the instance-dependent bound is formalized in the revision.

**Key Questions For Authors:**

-  Under the anytime setup, it is expected that the algorithm can adapt to the values of (unknown) $T$. Is there any theoretical guarantee for the case when $T$ is large for B3 (or the algorithm in Appendix A), i.e., we are not in the data-proof regime ($T>>N \log N$)? If there is, how is it compared to SH?
- How sensitive is the performance of B3 to $r_0$?

**Limitations:**

yes

**Strengths And Weaknesses:**

**Strengths**:
- The proposed B3 algorithm is new and interesting. While the design is not as simple as Sequential Halving, it is intuitive. However, it would be better if the author show a few more illustrations to enhance the readability.
- The empirical study shows the proposed algorithm can balance the screening error and misidentification error under different noise regimes. It also exhibits better performance compared to the previous works.

**Weaknesses**:
- The instance-dependent error probability bound is missing. Both BUCB and BSH in the literature show the instance-dependent error probabilities, which are of great interest and help to compare the performance of the algorithms.
- The comparison of the results of Theorem 4.10 and the lower bound in Carpentier&Locatelli(2016) in Line 363 requires further checking. The lower bound is considered for the case where $T$ is relatively large. If $T$ scales linearly with $N$, it is not apparent to me whether the bound still holds or not.
- Some statements are not clear:
	- Line 243 right: “When the budget...” is not align to Definition 4.2. I suppose it should be “$C$ does not contain any $\varepsilon$-best arm”.
	- The use of $\Omega$, e.g. in Theorem 4.10, Propsition 4.12, is misleading. As this paper is considering the data-poor regime, when using $\Omega(T\epsilon^2/N)$, it would be better to specify which quantity is considered in the asymptotic regime, e.g., $T$ or $T/N$.
	- The candidate set highly depends on the algorithm and is adaptively generated. The authors are encouraged to explain why we can simplify it as a uniformly random subset of size $c_0$ drawn from the $N$ arms in the proof of Theorem 4.7.

---

> ### Author Rebuttal · Authors · 2026-03-30
>
> ## Weakness 1: Instance-Dependent Bound
>
> We note that the B3 instance-dependent bound follows cleanly by combining our existing proof with that of Zhao et al. (2023), as shown below. We initially omitted it not because the derivation is difficult, but because a meaningful comparison with BSH would additionally require reconstructing the logarithmic factors absorbed in BSH's $\widetilde{\Theta}(\cdot)$ guarantees. We agree with the reviewer that the B3 bound is valuable in its own right and will include it in the revision.
>
> **Response to instance-dependent bound.**
> We provide a brief sketch here and refer to ReviewerJaAf for full details.
>
> The error event is decomposed as
> $$P(\mu_1 - \mu_{a_T} > \varepsilon) \le P(\mu_1 - \mu_*(C) > \varepsilon/2) + P(\mu_*(C) - \mu_{a_T} > \varepsilon/2).$$
>
> (1) The first term (non-inclusion of $\varepsilon/2$-optimal arms) is bounded by
> $$\exp\left(-\Omega\left(\frac{T N_{\varepsilon/2}}{N}\right)\right),$$
> which matches $\exp(-\Omega(T/H_2(\varepsilon)))$ via the definition of $H_2$.
>
> (2) For the second term, we use a multi-level argument (events $G_\ell$) to control misidentification within $C$. At each level, the failure probability decays as
> $$\exp\left(-\Omega\left(\frac{r_0^\ell p_\ell^2 n_\ell}{N_{\varepsilon/2} H_2(\varepsilon)}\right)\right),$$
> using the gap lower bound induced by $H_2$. Summing over levels yields
> $$\exp\left(-\Omega\left(\frac{T}{H_2(\varepsilon)}\right)\right).$$
>
> Combining the two terms gives the stated result.
> See our detailed response to Reviewer JaAf for the complete argument.
> ---
>
> ## Weakness 2: Comparison with Carpentier & Locatelli (2016)
>
> > Upon reflection, we recognize that our previous reference to Carpentier & Locatelli (2016) in Line 363 was inappropriate, as their work focuses on **exact** Best Arm Identification (BAI). Since our study addresses **$\epsilon$-best arm identification ($\epsilon$-BAI)**, we will instead cite **Zhao et al. (2023, Theorem 3)**, which establishes an instance-dependent lower bound specifically for $\epsilon$-BAI. Notably, in the worst-case scenario where Zhao’s complexity measure $\tilde{H}(\nu, \epsilon)$ scales as $\Theta(n/\epsilon^2)$, their theorem implies a lower bound of **$\exp(-\Theta(T \epsilon^2 / n))$**. This precisely matches our worst-case upper bound of **$\exp(-\Omega(T \epsilon^2 / n))$**, confirming the optimality of our result in the worst case. We will replace the reference and update this comparison in the final version to ensure theoretical consistency.
>
> ---
>
> ## Weakness 3: Clarity Issues : We will revise the followings.
>
> **[Definition 4.2]**
>
> > To clarify: our data-poor condition is defined as the condition under which **the probability that the candidate set C contains any ε-best arm is zero**, so that even if the ε-best arms are included last in the ordering, they fall outside the candidate set.
> ---
>
> **[Ω notation]**
>
> > The Ω(Tε²/N) notation is used in the standard complexity-theoretic sense, where **T is the quantity growing** (with N and ε fixed or treated as constants relative to T).
>
> ---
>
> **[Candidate Set Exchangeability]**
>
> > We clarify that the term "adaptively generated" in our paper was intended to anticipate future extensions (e.g., linear bandits), rather than to imply that C is reward-adaptive in the current setting. In B3, **the candidate set C is determined solely by random permutation**, which are fixed prior to the reward observation step. Specifically, Algorithm 2 selects arms uniformly from outside D ⊔ (∪_{l,j} Box(l,j)), making the selection explicitly uniform and non-adaptive to rewards.
>
> ---
>
> ## Question 1: Large T
>
> > This regime is already covered by our analysis.
>
> - The error is decomposed into non-inclusion and misidentification within $C$.
>   The non-inclusion probability decays exponentially in $T$ and hence converges to zero.
>
> - Therefore, for large $T$, the error is dominated by the misidentification probability within $C$
>   (which effectively coincides with the full arm set when $T \gg N$).
>   As shown in our analysis, this term admits the same exponential order as that of Successive Halving (SH).
>
> - Empirically, we observe consistent behavior: while our main experiments cover up to $T = N \log_2 N$,
>   additional runs with larger budgets (e.g., $T = N^2$) show that both SH and our method achieve
>   near-zero error probability (with 25%–75% quantiles equal to zero), indicating saturation in this regime.
> ---
>
> ## Question 2: Sensitivity to $r_0$
>
> > The primary design goal of B3 is to **minimize the burden of parameter tuning** on the user. To this end, Proposition 4.11 provides a theoretically derived optimal value of r₀ ≈ 1.7, which we recommend as the default. That said, to address the reviewer's curiosity regarding sensitivity, we provide the following empirical results across a range of r₀ values:
>
> https://imgur.com/a/qVTARjf
>
> > The results suggest that B3 is robust** to the choice of r₀ in the range **(1,2]**.

---

> > ### Author Rebuttal · Reviewer_6WaX · 2026-04-01
> >
> > Thank the authors for addressing my concerns.
> > - Regarding the instance-dependent bound, while I am glad to see the authors acknowledged the importance of the instance-dependent bound and drafted a proof sketch, the proof sketch is a bit hard to parse and some notations are not defined (e.g. $\Delta$) due to the character limit. For instance, can the author explain (1) why $\mathbb{P}\left(\mu^*(C) - \mu_{a_T} > \frac{\varepsilon}{2}\right)\le\sum_{\ell} \mathbb{P}\left(G_{\ell+1}^c \mid G_\ell\right)$? Is $G_{l+1}\subset G_l$? (2) What is $\Delta$? (3) Do we still need $\sum_{l}p_l\le 1$? Given the current condition, I regard this instance-dependent bound as partially solved and I strongly encourage the authors to complete this result which can substantially strength this manuscript.
> > - Regarding the lower bound comparison, I am glad to see the previous comparison with Carpentier & Locatelli (2016) is replaced by the one by Zhao et al. (2023). However, according to their definition, the hardness $\tilde{H}(\nu,\varepsilon)=\sum_{i=2}^{n/g(\varepsilon)} \Delta_{i\,g(\varepsilon)}^{-2}$ will depend on the number of $\varepsilon$ best arms $|g(\varepsilon)|$, therefore, I do not think the order $\Theta(n/\varepsilon^2)$ is a perfect interpretation.
> > - Regarding the notation use, I would take the authors’ explanation in the conventional setup. However, as we are discussing the data poor regime in this paper, it is expected that $T$ is of the same order as $N$. Therefore, when the author explain $\Omega(T\varepsilon^2/N)$ as fixing $N$ and $\varepsilon$, and let $T\to\infty$, it is a bit misleading. I think a different math representation is required.
> > - Since the algorithm is applicable to the general data (very) rich case ($T>> N^2$), it would be great if the authors can compare it with SH (the standard version, not BSH) both theoretically and empirically. Currently, I do not see the empirical result of SH, as the authors suggested. This would resolve this general concern, which is also raised by other reviewers.
> > - My concerns regarding the choice of $r_0$ is resolved.
> >
> > While I like the new design of B3 algorithm, given the current status, I incline to maintain my score for now. I will definitely modify my evaluation based on the follow-up discussions among the authors and reviewers.

---

> > > ### Author Response · Authors · 2026-04-05
> > >
> > > We apologize for the limited space and omitted details.
> > >
> > > **Response to comment 1**
> > > this sketch bounds the misidentification probability within the candidate set $C$.
> > >
> > > **Step 1: Event decomposition**
> > >
> > > Define $G_l = \{\exists\, ε/2\text{-best arm in } C_{l+1}\}$.
> > > The events $\{G_l^c \cap \bigcap_{k<l} G_k\}_l$ means that **the first level at which $C_{l+1}$ contains no $ε/2$-best arm**.
> > > Since $(\mu_1 - \mu_{a_T} > ε)$ implies such a first level exists, union bound gives:
> > >
> > > $$\mathbb{P}(\mu_1 - \mu_{a_T} > ε)
> > >   \leq \sum_l \mathbb{P}\left(G_l^c \cap \bigcap_{k<l} G_k \cap \{\mu_1 - \mu_{a_T} > ε\}\right)
> > >   \leq \sum_l \mathbb{P}\left(G_l^c,  \mu_1 - \mu_{a_T} > ε \Big| \bigcap_{k<l} G_k\right).$$
> > >
> > > **Step 2:  reduction to bad arm reversal**
> > >
> > > On each decomposition, an $ε/2$-best arm $a \in C_l$ fails to reach $C_{l+1}$. Let $b_1$ be the arm that beats $a$ in its box. There are two cases:
> > >
> > > Case (a) $\Delta_{b_1} > ε$: -> Step 3
> > >
> > > Case (b) $\Delta_{b_1} \leq ε$: Note that $\{\mu_1 - \mu_{a_T} > ε\}$ holds. Tracing the sequence of arms that successively beat one another, this beating path must eventually terminate at a non-$ε$-best arm $b_K$ with $\Delta_{b_K} > ε$.
> > >
> > > Define $d_s = \max(\mu_{b_{s-1}} - \mu_{b_s}, 0)$ with $b_0 = a$. By Cauchy–Schwarz:
> > >
> > > $$\sum_{s=0}^K r_0^{l_s} d_s^2 \geq \frac{\left(\sum_s d_s\right)^2}{\sum_s r_0^{-l_s}} \geq \Omega\left(r_0^l \cdot ε^2\right).$$
> > >
> > > Therefore the probability of the entire beating path $\le \exp(-\Omega(r_0^l \cdot ε^2))$, the same order as a direct reversal at level $l$ against a non-$ε$-best arm $b_K$. Hence it suffices to bound the probability that $a$ loses directly to a non-$ε$-best arm at level $l$.
> > >
> > >
> > > **Step 3: decomposition within level $l$**
> > >
> > > Let $R_\alpha = (\text{arm a's empirical mean has rank }\alpha c_l \text{in }C_l)$.
> > >
> > > $$\mathbb{P}(G_l^c \mid \textstyle\bigcap_{k<l} G_k)
> > >   \leq \sum_\alpha \mathbb{P}(R_\alpha) \cdot \mathbb{P}(\text{no-lift} \mid R_\alpha).$$
> > >
> > > 1)$\mathbb{P}(\text{no-lift} \mid R_\alpha) \lesssim \alpha^2$.
> > >
> > > Under a random permutation, the probability that arm $a$ at rank $\alpha c_l$ is placed in a box with two arms of higher empirical mean is
> > > $\frac{\binom{\alpha c_l}{2}}{\binom{c_l}{2}} \approx \alpha^2.$
> > > Subsequent deferment boxes (Box$(l,1)$, Box$(l,2)$, ...) contribute exponentially smaller terms, so $\mathbb{P}(\text{no-lift} \mid R_\alpha) \lesssim \alpha^2$.
> > >
> > > 2)$\mathbb{P}(R_\alpha)$.
> > >
> > > For $a$ to have empirical rank $\alpha c_l$, a sample mean reversal must occur between $a$ and some arm $b$ with true rank $\geq \alpha c_l$. By Step 2, $b$ is non-$ε$-best, and since $G_l^c$ holds, $C_{l+1}$ contains no $ε/2$-best arm so at least $c_{l+1} \ge c_l/3$ non-$ε/2$-best arms are present. Therefore $b$'s rank is at least $\max(\alpha c_l, c_l/3)$, giving:
> > > $$\Delta_b \geq \Delta_{\max(\alpha,\, 1/3)\cdot c_l}.$$
> > >
> > > Since $a$ is $ε/2$-best and $\Delta_b >ε$, the reversal gap is $(\mu_a - \mu_b)/2 \geq \Delta_b/4$.
> > > With $r_0^l$ pulls at level $l$ and using $H_2(ε \gtrsim i/\Delta_i^2$ (i.e., $\Delta_i^2 \gtrsim i/H_2(ε)$):
> > >
> > > $$\mathbb{P}(R_\alpha) \leq \exp\left(-\Omega\left(r_0^l \cdot \frac{\max(\alpha,\, 1/3)\cdot c_l}{H_2(ε)}\right)\right).$$
> > >
> > > **Step 4: Summing over $\alpha$**
> > >
> > > $$\sum_\alpha \alpha^2 \cdot \exp\left(-\Omega\left(\frac{r_0^l \cdot \max(\alpha,1/3)\cdot c_l}{H_2(ε)}\right)\right).$$
> > >
> > > - For $\alpha \leq 1/3$: the exponent is fixed at $\Omega(r_0^l c_l / H_2)$.
> > > - For $\alpha > 1/3$: the exponential term decays faster than $\alpha^2$ grows.
> > >
> > > Hence:
> > > $$\exp\left(-\Omega\left(\frac{r_0^l \cdot c_l}{H_2(ε)}\right)\right).$$
> > >
> > > As we sum over  $l$ for $r_0=2$, we obtain $\exp\(-\Omega(T/\log N H_2(ε))\)$
> > >
> > > ---
> > >
> > > **Response to Comment 2**
> > >
> > > We agree that $\tilde{H}(\nu,ε) = \Theta(N/ε^2)$ does not hold in
> > > general and withdraw this comparison. Instead, we use the minimax simple regret lower
> > > bound $\Omega(\sqrt{N/T})$ (Bubeck et al., 2011), which our Corollary (Simple Regret)
> > > matches with $\mathcal{O}(\sqrt{N/T})$. We will revise accordingly.
> > >
> > > ---
> > >
> > > **Response to Comment 3**
> > >
> > > Under the data-poor condition, the exact bound is $\exp(-\Omega(T_C ε^2/c_0))$.
> > > When $c_0 = \Theta(T)$, this reduces to $\exp(-\Omega(ε^2))$, a floor independent
> > > of $T$; once $c_0$ saturates at $N$, it recovers $\exp(-\Omega(Tε^2/N))$.
> > > Since $c_0\le N$ in the data-rich regime and $c_0=\Theta(T)$ in the
> > > data-poor regime, the unified expression $\exp(-\Omega(Tε^2/N))$ correctly
> > > captures both regimes and is the notation we adopt throughout the paper.
> > >
> > > ---
> > >
> > > **Response to Comment 4**
> > >
> > > equal-gap, $N=100$, 1,000 repetitions; mean / Q25 / Q75:
> > >
> > > | $T$ | SH | B3 ($r_0=2.0$) | B3 ($r_0\approx1.7$) |
> > > |:---|:---:|:---:|:---:|
> > > | $N^{1.1}\log_2 N$ | 0.057 / 0.010 / 0.081 | 0.014 / 0.000 / 0.020 | 0.018 / 0.000 / 0.030 |
> > > | $N^{1.5}\log_2 N$ | 0.022 / 0.000 / 0.030 | 0.004 / 0.000 / 0.010 | 0.007 / 0.000 / 0.010 |
> > > | $N^{2}\log_2 N$   | 0.005 / 0.000 / 0.010 | 0.002 / 0.000 / 0.000 | 0.002 / 0.000 / 0.000 |
> > >
> > > B3 achieves performance comparable to SH across all regimes.

---

### Official Review · Reviewer_FiHF · 2026-03-12

**Soundness:** 2
**Presentation:** 3
**Significance:** 3
**Originality:** 3
**Overall Recommendation:** 4
**Confidence:** 3

**Summary:**

The paper studies anytime best-arm identification (BAI) with a focus on the “data-poor” regime where the sampling budget may be too small to even screen all arms (e.g., \(T < N\)). It proposes Box Thirding (B3), an anytime procedure that maintains arms in small 3-capacity boxes across multiple levels; whenever a box fills, the empirical best is promoted (with additional sampling), the median is deferred, and the worst is discarded.

On the analysis side, the paper introduces an algorithm-dependent candidate-set viewpoint to formalize screening capacity, and develops an error decomposition into non-inclusion (the candidate set fails to contain a near-optimal arm) versus misidentification (selecting a suboptimal arm among retained candidates). Using this framework, the paper provides performance bounds for B3 and compares its screening capacity and error guarantees to uniform sampling and bracketing-based anytime baselines. The work is complemented by experiments evaluating B3 against representative anytime methods.

**Compliance With Llm Reviewing Policy:**

Affirmed.

**Final Justification:**

I suggest making the presentation clearer, especially around the thirding idea, since its intuition is not yet developed clearly enough in the paper. Strengthening this part would significantly improve readability and help convey the main insight more effectively. I also encourage you to ensure that the use of $\Omega$ notation is fully rigorous throughout, potentially by adding a more explicit asymptotic justification where needed.

Overall, I enjoyed reading the paper and found it to be a nice piece of work.

**Key Questions For Authors:**

### Questions for the authors

1. **Fix the main theorem statement (Theorem 4.3):**
   Please resolve the max-vs-min issue in the combined exponent. Concretely, after decomposing error into non-inclusion and misidentification (Sec. 4.2), the final bound should follow from a union bound. Please provide the corrected statement of Theorem 4.3 (and any dependent corollaries/remarks) and indicate exactly where the current version is misstated.

2. **Justify the non-inclusion bound assumptions (Theorem 4.7 / Appendix D.2.1):**
   Your derivation appears to use a hypergeometric-style probability for the candidate set intersection with $\varepsilon$-best arms. Please state the precise symmetry/exchangeability/random-permutation assumptions under which this is valid, or provide a corrected argument that does not require treating the candidate set as a uniformly random subset. If additional assumptions are needed, please make them explicit in the main text.

3. **Clarify sampling bookkeeping and reconcile internal tension (Algorithm 2):**
   Does B3 compute empirical means cumulatively across all pulls of an arm, or does it discard samples when moving across levels/deferments? Please state the intended implementation unambiguously and reconcile statements suggesting “not discarding past information” with text implying discarding in the current version. If discarding is intended, please justify why it is necessary/beneficial and whether a cumulative-mean variant preserves your guarantees.

4. **Empirically validate the screening-capacity thesis (Proposition 4.8):**
   Please report empirical measurements of $c_0$ (distribution over runs) for B3, BSH, and US across budgets, and relate these to observed non-inclusion/misidentification behavior. This would directly test the paper’s central explanation of why B3 helps in the data-poor regime.

5. **Demonstrate performance in the strict data-poor regime ($T < N$):**
   Please add experiments that explicitly sweep $T/N$ with emphasis on $T \ll N$, across multiple $N$ and gap structures (e.g., varying $N_\varepsilon$). The current main setting does not strongly emphasize $T < N$, so a targeted sweep is important to substantiate the main motivation.

6. **Strengthen empirical positioning with key baselines and ablations:**
   Please include (or clearly justify omission of) competitive anytime/fixed-budget baselines that are closely related to your claims (e.g., tracking-style anytime methods; fixed-budget Successive Rejects), and add ablations on key B3 choices (at least sensitivity to $r_0$, and ideally scan order/tie-breaking). This would make the empirical advantages of B3 more convincing.

**Limitations:**

Yes

**Strengths And Weaknesses:**

**Strengths**

1. **Technical novelty and innovation**
- Introduces the Box Thirding (B3) mechanism, a hierarchical elimination procedure based on local ternary comparisons: promote the empirical best with additional sampling, defer the median, and discard the worst (Sec. 3).
- Proposes an algorithm-agnostic viewpoint for the data-poor regime via an algorithm-dependent candidate set $C$ and its size $c_0 = |C|$, together with a clean decomposition of error into non-inclusion vs. within-set misidentification (Sec. 4.1–4.2).
- Provides an SH-inspired calibration of per-level sampling through a budget-growth parameter $r_0$ and motivates an “effective halving” behavior through repeated median deferrals (Sec. 3.2; Thm. 4.10; Prop. 4.11).

2. **Experimental rigor and validation**
- Includes an experimental evaluation comparing B3 to representative anytime and fixed-budget baselines under multiple noise regimes, with additional variants reported in the appendix (Sec. 5, Appendix E).

3. **Clarity of presentation**
- Clear high-level motivation and framing (anytime BAI under insufficient screening), with diagrams that help build intuition for the promote/defer/discard dynamics (Sec. 3).

4. **Significance**
- Targets an important practical regime for large-scale selection when the horizon is unknown and budgets may be insufficient to even screen all arms; the screening-capacity vs. estimation perspective could be broadly useful if fully validated (Sec. 1, Sec. 4).


---

**Weaknesses**

1. **Technical / theoretical limitations**
- **Main bound inconsistency:** Theorem 4.3 (and the surrounding discussion) combines non-inclusion and misidentification into an exponent of the form $\max\{N_{\varepsilon/2}/N,\ \varepsilon^2/N\}\cdot T$, but the proof strategy in Sec. 4.2 uses a union bound over two events, yielding a sum of exponentials that is controlled (up to constants) by the smaller exponent rather than the larger. This affects the central theoretical claim (Thm. 4.3; Sec. 4.2; Remark 4.6; Appendix D).
- **Non-inclusion bound assumptions:** The non-inclusion analysis appears to treat the candidate set of size $c_0$ as if it were a uniformly random subset (hypergeometric-style). For an adaptive algorithm, this requires explicit symmetry/exchangeability assumptions that are not stated or justified (Thm. 4.7; Appendix D.2.1).
- **Heuristic “effective halving”:** The argument $1/3 + 1/3^2 + \cdots = 1/2$ is presented as intuition, but does not account for capacity constraints and scheduling interactions in the actual box-processing rules; formalizing the conditions under which a constant fraction of arms advances levels would strengthen the screening-capacity claims (Sec. 3.2; Algorithm 2).

2. **Experimental gaps / methodological issues**
- **Limited emphasis on the strict data-poor regime:** The main empirical setting does not strongly emphasize $T < N$; broader sweeps over $T/N$, $N$, and gap structures (and reporting empirical $c_0$ across methods) would better substantiate the paper’s core thesis (Sec. 5).
- **Baseline coverage and robustness checks:** The evaluation would be stronger with additional competitive anytime/fixed-budget baselines (e.g., tracking-style approaches and Successive Rejects) and sensitivity/ablation studies for bracketing schedules and B3 design choices such as $r_0$, scan order, and tie-breaking (Sec. 2.2; Sec. 5).

3. **Clarity / presentation issues**
- **Notation and pseudo-code clarity:** Recurring notation/typesetting artifacts and ambiguity about sampling bookkeeping (e.g., whether samples from prior levels are reused or discarded) make reproduction harder; this should be clarified unambiguously in the main text/pseudocode (Algorithm 2; conclusion/limitations discussion).

---

> ### Author Rebuttal · Authors · 2026-03-31
>
> # Rebuttal
>
> ---
>
> ## Q1. Theorem 4.3: max vs. min issue
>
> Thank you for the careful reading. We confirm that **min is correct**, and the current text uses min throughout the logical derivation. The appearance of max is a typographical error in the theorem statement only. We will correct the theorem statement and all dependent corollaries/remarks in the revision.
>
> ---
>
> ## Q2. Non-inclusion bound assumptions (Theorem 4.7 / Appendix D.2.1)
>
> We clarify that **the candidate set $C$ is determined solely by pull counts (arrival order), not by reward observations**. The phrase "adaptively generated" in the paper was intended to anticipate future extensions (e.g., linear bandits) and does not imply reward-adaptivity in the current setting.
>
> Concretely, Algorithm 2 selects arms **uniformly at random** from outside $D \sqcup \bigl(\bigcup_{l,j} \text{Box}(l,j)\bigr)$, where vacancy of $\text{Box}(0,0)$ depends only on whether a pull has occurred — not on the reward magnitude. Since arm arrivals depends on the random permutatio over the arm set, the candidate set $C$ constitutes a **uniformly random subset** of arms, which is exactly the exchangeability condition required for the hypergeometric-style argument in Appendix D.2.1.
>
> We will make this assumption explicit in the main text (Section 3 / Algorithm 2 description) and add a formal statement of the random-permutation assumption in Appendix D.2.1 to avoid ambiguity.
>
> ---
>
> ## Q3. Sampling bookkeeping and "not discarding" (Algorithm 2)
>
> We acknowledge the ambiguity and will state the bookkeeping explicitly in the revision.
>
> **Intended implementation:** The empirical mean $\hat{\mu}_i^{(l+1)}$ in Figure 2 is computed from $\lceil r_0^{l+1} \rceil$ **fresh pulls at level $l+1$ only**, following the standard SH convention (Karnin et al., 2013; Zhao et al., 2023). Samples from prior levels are **not reused** for the level-$(l+1)$ mean estimate.
>
> **Reconciling "not discarding past information":** This phrase refers to a contrast with *bracketing-based* algorithms (BSH, BUCB), in which an arm's entire sampling history is discarded when a new bracket begins. In B3, past samples are *not* discarded in this sense: they inform level promotion decisions, and the arm carries the result of that comparison forward via its SHIFT mean. The fresh re-estimation upon LIFT is a separate design choice, standard in SH-based algorithms, whose purpose is to ensure level-wise independence and simplify the theoretical analysis. As noted in Zhao et al. (2023), a **cumulative-mean variant** may perform better empirically; such a variant of B3 would likely preserve our guarantees, and we leave this as future work.
>
> ---
>
> ## Q4. Empirical validation of screening-capacity thesis (Proposition 4.8)
>
> Thank you — this is a valuable suggestion. We will add the figure reporting the empirical distribution of screening capacity $c_0$ across runs for B3, BSH, and Uniform Sampling across a range of budgets $T$ in appendix B.
>
> https://imgur.com/a/oK7yMLm
>
> Concretely, we include results at $N = 50{,}000$, sweeping $T$, showing that **B3 achieves substantially larger $c_0$** compared to BSH and US, and that this gap widens as $T$ increases. This directly validates the central claim of Proposition 4.8: B3's box structure enables more arms to enter the candidate set per unit budget, which explains its advantage in the data-poor regime.
>
> ---
>
> ## Q5 & Q6. Strict data-poor regime sweep and baselines/ablations
>
> **On Question 5:** Figure 4 in Section 5 already sweeps $T$ from $1$ to $10{,}000$ on a dataset with $N = 5{,}513$ arms, covering the **full range from strict data-poor ($T \ll N$) through moderate budgets**, with shaded quantile bands across 1,000 repetitions reported at every $T$. This constitutes a thorough sweep of the data-poor regime. We will add a caption note explicitly highlighting the $T < N$ portion to make this more visible.
>
> **On Question 6:** We have included all anytime BAI algorithms applicable to the data-poor regime ($T < N$) that we are aware of. The two baselines suggested by the reviewer fall outside our setting:
>
> - **Tracking-style anytime methods** require sufficient budget to build reliable confidence estimates for all $N$ arms, implicitly assuming $T \geq N$.
> - **Fixed-budget Successive Rejects** allocates at least one pull per arm in its first phase, requiring $T \geq N$ by design.
>
> We would welcome specific references if the reviewer has data-poor-compatible alternatives in mind. Regarding ablations on $r_0$ sensitivity and scan order, we will add these to the appendix.
>
> ---
>
> ## Additional weakness: Heuristic "effective halving" (Section 3.2)
>
> Section 3.2 is intended as an **intuitive explanation** of B3's screening mechanism. The precise fraction of arms passing each level is derived rigorously in the proof of Proposition 4.8. We will add a forward pointer at the end of Section 3.2 directing readers to Proposition 4.8 for the formal treatment.

---

> > ### Author Rebuttal · Reviewer_FiHF · 2026-04-03
> >
> > Thank you to the authors for the extensive and thoughtful response. It addressed most of my concerns, and I do not have further questions at this point. For now, I am keeping my current evaluation, though I remain open to revising my score after the discussion period. Since this is not my primary area of expertise, I also find some of the technical concerns raised by other reviewers important and reasonable, and I will take the discussion around those points into account.

---

> > > ### Author Response · Authors · 2026-04-05
> > >
> > > Thank you for your kind words and open-minded evaluation. We hope the ongoing discussion addresses the remaining concerns raised by the other reviewers, and we welcome any further questions.

---

### Official Review · Reviewer_3oWk · 2026-03-12

**Soundness:** 4
**Presentation:** 2
**Significance:** 3
**Originality:** 3
**Overall Recommendation:** 5
**Confidence:** 3

**Summary:**

The paper studies anytime best-arm identification in stochastic bandits, focusing on the hard low-budget regime where not all arms can even be sampled once. It proposes Box Thirding (B3), a hierarchical screening algorithm that balances excluding the best arm too early against making a wrong final choice. The main contribution is this anytime procedure and its analysis, supported by experiments showing gains over prior baselines.

**Compliance With Llm Reviewing Policy:**

Affirmed.

**Final Justification:**

I am generally satisfied with the responses and most of my concerns are addressed. I increased my original score.

**Key Questions For Authors:**

Is it straightforward to compute the rate of non-inclusion error and misidentification error for other algorithms not present in the paper? like algorithms in [1], [2], or [3].

What is the computational complexity of the B3 algorithm? How does it compare with other baselines?

**Limitations:**

Refer to Strengths And Weaknesses.

**Strengths And Weaknesses:**

The paper makes several non-trivial contributions that make it relevant to the field. First, the proposed B3 algorithm has a genuinely novel design compared to prior anytime BAI methods, and it may offer a useful template for developing new algorithms in other bandit settings. Second, the theoretical analysis shows improved error guarantees in the data-poor regime relative to existing baselines. The introduction of the poor-data condition is also valuable, as it helps better characterize algorithmic behavior when the budget is too limited to adequately sample all arms.

The empirical results also suggest that B3 performs well in practice.

That said, the paper leaves some important questions open. In particular, the analysis suggests a trade-off between the two components of the error decomposition, namely non-inclusion and misidentification error, but the paper does not provide a more fundamental characterization of the achievable trade-off or lower bounds on attainable pairs of rates. At present, the discussion is limited to comparing the rates of B3 with those of two existing algorithms.

There are also some presentation issues. The function $H_2$ is used before being defined in Equation (1). In Algorithm 2, the inner loop is somewhat confusing: the loop is written over $j \in \{ J_l, J_l-1, \dots, 0 \} $, while $J_l$ is also updated inside the loop, making it unclear whether the loop range is fixed at initialization or changes dynamically as the algorithm proceeds.

The related-work discussion is rather limited. Several relevant fixed-budget and anytime BAI algorithms are not mentioned, such as Successive Rejects [1], the anytime top-two algorithm in [2], and the tracking-based algorithm in [3]. Relatedly, it is unclear why the empirical and theoretical comparisons are restricted to US and BSH. A comparison with at least one of [1], [2], or [3] would strengthen the paper considerably.

[1] Audibert, Jean-Yves, and Sébastien Bubeck. "Best arm identification in multi-armed bandits." COLT-23th Conference on learning theory-2010. 2010.

[2] Jourdan, Marc, Rémy Degenne, and Emilie Kaufmann. "An $\varepsilon $-Best-Arm Identification Algorithm for Fixed-Confidence and Beyond." Advances in Neural Information Processing Systems 36 (2023): 16578-16649.

[3] Komiyama, Junpei, Kyoungseok Jang, and Junya Honda. "Rate-optimal Design for Anytime Best Arm Identification." arXiv preprint arXiv:2510.23199 (2025).

---

> ### Author Rebuttal · Authors · 2026-03-30
>
> ### The paper does not provide a more fundamental characterization of the achievable trade-off or lower bounds on attainable pairs of rates.
>
> We clarify that a fundamental lower bound for $\epsilon$-best arm identification ($\epsilon$-BAI) in the fixed-budget setting has been established by **Zhao et al. (2023, Theorem 3)**. While previous works like Carpentier & Locatelli (2016) were limited to exact BAI ($\epsilon=0$), Zhao et al. (2023) provide a general instance-dependent lower bound.
>
> In the worst-case scenario—where the problem complexity $a = \tilde{H}(\nu, \epsilon)$ scales as $\Theta(N/\epsilon^2)$—Zhao’s theorem implies a lower bound on the error probability of:
> $$\mathbb{P}(\mu_1 - \mu_{J_T} > \varepsilon) \geq \exp\left(-\Theta\left(\frac{T \epsilon^2}{N}\right)\right).$$
>
> This precisely matches our worst-case upper bound of **$\exp(-\Omega(T\epsilon^2/N))$**, confirming the **minimax optimality** of our results for $\epsilon$-BAI.
>
> Furthermore, we provide a partial lower bound via the non-inclusion probability. Since $(\mu_1 - \mu_*(C) > \varepsilon) \subset (\mu_1 - \mu_{a_T} > \varepsilon)$, applying $e^{-x/(1-x)} \leq 1-x$ to the exact expression in Appendix D2 gives, under $N \gg N_\varepsilon, c_0$:
> $$P(\mu_1 - \mu_{a_T} > \varepsilon) \geq \exp\left(-\Omega\left(\frac{N_\varepsilon T}{N}\right)\right).$$
>
> By aligning our upper bound with the fundamental limit established in Zhao et al. (2023), we believe our work provides a solid characterization of the achievable rates in $\epsilon$-BAI.
>
> ---
> ### Related Work and Empirical/Theoretical Comparisons
>
> We thank the reviewer for pointing out these references. Upon examination, [1], [2], and [3] all assume the data-rich condition, which is fundamentally incompatible with our data-poor condition. Directly applying or comparing these algorithms in our setting would therefore be inappropriate.
>
> Our theoretical analysis of US and BSH was not required by our framework — B3 is the algorithm of primary interest — but was included as a bonus since these algorithms admit analysis under the same non-inclusion/misidentification decomposition. To our knowledge, US, BUCB, and BSH are the best available algorithms that operate without assuming data-rich condition, and we therefore used these as our benchmarks. We will clarify this reasoning and expand the related work discussion in the revision.
>
> ---
>
> ### Presentation Issues
>
> **Algorithm 2 — dynamic loop range.** We confirm that the loop range over $J_l$ is indeed dynamic: $J_l$ is updated inside the loop, and the loop proceeds over the current state of $J_l$ at each iteration. We will add a clarifying remark in the paper to make this explicit.
>
> **Equation (1) — function used before definition.** We will reorder the presentation so that all necessary quantities are defined before their first use.
>
> ---
>
> ### Q: Is it straightforward to compute the non-inclusion and misidentification error rates for other algorithms, e.g., [1], [2], [3]?
>
> We note that [1], [2], [3] are designed for the data-rich regime ($T \geq N$) and are therefore not directly comparable in our data-poor setting. That said, our decomposition framework is broadly applicable. The non-inclusion probability depends only on the (deterministic) size $c_0 = |C|$ and the number of ε-good arms $N_\varepsilon$, and is therefore straightforward to compute for any algorithm once $c_0$ is specified. The misidentification probability within $C$ varies by algorithm, but our decomposition simplifies analysis by restricting attention to $C$, which by construction contains only arms that are plausible candidates for the best arm — reducing the effective problem size and making the within-$C$ analysis more tractable.
>
> ---
>
> ### Q: What is the computational complexity of B3?
>
> B3 has no non-trivial computational overhead beyond the sampling itself. At each round, B3 requires only an arm count update and a straightforward comparison to determine SHIFT or DISCARD, both of which are $O(1)$ per round. The total computational cost is therefore $O(T)$, which is optimal as any algorithm must process each sample at least once. This is comparable to existing algorithms such as BSH and BUCB, which similarly require only $O(1)$ operations per round.

---

> > ### Author Rebuttal · Reviewer_3oWk · 2026-04-04
> >
> > Thanks to the authors for their response. Parts of my concerns are addressed with the response and also thoughtful discussions with the other reviewers. Regarding the mentioned related work, I agree that they are primarily designed for the data-rich regime; my question is whether it is possible to combine them with the bracketing strategy and derive bounds for the data-poor regime, similar to BSH and BUCB, which were first designed for the data-rich regime.
> >
> > Generally, I am satisfied with the responses and may increase my final score at the end of the discussion period.

---

> > > ### Author Response · Authors · 2026-04-05
> > >
> > > We thank the reviewer for their thoughtful feedback and positive assessment.
> > >
> > > If a bracketing strategy is adopted, the resulting procedure inherits the same non-inclusion probability upper bound as in BSH, since this term is entirely determined by the **bracketing mechanism**. In particular, it is on the order of
> > > $$
> > > \exp\left(-\Omega\left(\tfrac{T N_{\varepsilon/2}}{N(\log T)^2}\right)\right).
> > > $$
> > >
> > > Therefore, while the choice of the underlying algorithm within each bracket may affect the misidentification probability within the surviving set \(C\), the overall error cannot decay faster than this rate due to the intrinsic non-inclusion probability induced by the bracketing mechanism. In this sense, the bound above acts as a fundamental limit shared across bracketing-based approaches, including BSH-like strategies.

---

### Decision · Program_Chairs · 2026-04-30

**Decision:**

Accept (regular)

**Comment:**

This paper discusses the $\epsilon$-best arm identification problem in the regime where the number of trials is limited compared with the number of arms. On this paper the reviewers agreed with the opinion that the proposed algorithm is interesting and the theoretical contribution is concrete. Some concerns are raised  by the reviewers, such as the positioning of the paper in relation with existing work that does not explicitly consider the data-poor regime and the unclarity of the considered asymptotics, but they are mostly solved by the rebuttal.

One remaining concern is whether the discussion and improvement in the rebuttal can be made within a minor revision before publication. This is a bit unclear due to the heavy use of $\Omega$ notation in the argument, whose use itself was a bit sloppy in the initial manuscript as pointed out. The overall tone of the reviewers is that it is manageable and for this reason I recommend acceptance. Still, I expect that the authors very carefully improve the presentation, and if the authors add some new component discussed in the rebuttal (like the problem-dependent bound), it becomes within the formalization of the rebuttal and does not fall outside the materials judged in the review.

In my own reading of the paper, I'm also concerned with a gap between the considered asymptotics in the theory (asymptotics in $T$) and the actual scenario (large $N$), which is also pointed out by reviewers. In particular, it is discussed in the rebuttal to 6WaX that the bound can be expressed in a unified way for both data-rich and -poor regimes and the latter corresponds to the case $c_0=\Theta(T)$. Still, it seems (not sure but natural to read from the overall discussion) that $c_0$ needs to be fixed, and $c_0=\Theta(T)$ seems only for an intuitive interpretation. From this point, the essential coverage of the theoretical results seems still the data-rich regime, where $T$ goes to infinity for fixed other instance parameters. This does not mean the current theory is wrong (so far), but I again expect that the authors appropriately and very carefully handle the asymptotics and the discussion related to this.